# Geometric Logit Decoupling for Energy-Based Graph Out-of-distribution Detection

**Min Wang, Hao Yang, Qing Cheng,**[*] **Jincai Huang**[*]
National University of Defense Technology
{wangminwm, yanghao, huangjincai}@nudt.edu.cn;
sgggps@163.com

## Abstract

GNNs have achieved remarkable performance across a range of tasks, but their reliability under distribution shifts remains a significant challenge. In particular, energy-based OOD detection methods—which compute energy scores from GNN logits—suffer from unstable performance due to a fundamental coupling between the norm and direction of node embeddings. Our analysis reveals that this coupling leads to systematic misclassification of high-norm OOD samples and hinders reliable ID–OOD separation. Interestingly, GNNs also exhibit a desirable inductive bias known as angular clustering, where embeddings of the same class align in direction. Motivated by these observations, we propose GeoEnergy (Geometric Logit Decoupling for Energy-Based OOD Detection), a plug-and-play framework that enforces hyperspherical logit geometry by normalizing class weights while preserving embedding norms. This decoupling yields more structured energy distributions, sharper intra-class alignment, and improved calibration. GeoEnergy can be integrated into existing energy-based GNNs without retraining or architectural modification. Extensive experiments demonstrate that GeoEnergy consistently improves OOD detection performance and confidence reliability across various benchmarks and distribution shifts.

## 1 Introduction

Graph Neural Networks (GNNs) [1, 2] have achieved remarkable success in diverse applications, including social network analysis [3], drug discovery [4, 5], and traffic forecasting [6, 7]. However, their effectiveness relies on the assumption that training and testing data are independently and identically distributed *(i.i.d.)*, an assumption that often fails in real-world scenarios. When deployed in open environments, GNNs frequently encounter out-of-distribution (OOD) inputs [8] —samples that differ significantly from the training distribution. The inherent structural dependencies in graph-structured data further complicate OOD detection, as node relationships can propagate distributional shifts across the graph. Consequently, conventional GNNs often produce overconfident yet incorrect predictions for OOD samples [9, 10], undermining model reliability in safety-critical applications. To address this challenge, a growing body of research has explored OOD detection in graph-structured data. Generative models like GraphDE [11] model uncertainty via variational inference, while Bayesian methods such as GPN [12] propagate uncertainty using posterior distributions. More recently, energy-based methods [13, 14, 15] have gained attention for their simplicity and post-hoc applicability. These approaches define energy scores based on the logit outputs of GNNs and flag high-energy samples as potential OOD nodes. However, our analysis reveals that the effectiveness of energy scores hinges critically on the geometric structure of the logits themselves.

---

[*]Corresponding author.

39th Conference on Neural Information Processing Systems (NeurIPS 2025).

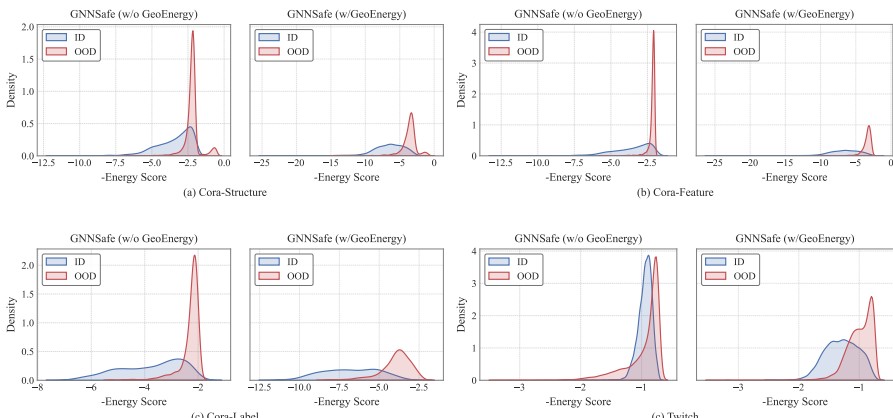

Figure 1: **Energy distributions under GNNSafe with and without GeoEnergy.** Across different graph shifts, integrating GeoEnergy significantly improves ID-OOD energy separation and stabilizes score distributions, leading to more reliable OOD detection.

Specifically, current energy-based methods typically compute the logit for class $j$ as an inner product: $f_j = \boldsymbol{W}_j^\top \boldsymbol{h}_i$, where $\boldsymbol{W}_j$ is the class weight vector and $\boldsymbol{h}_i$ is the node embedding. This formulation implicitly incorporates both the direction and the magnitude of $\boldsymbol{h}_i$. In graph settings, however, the norm $\|\boldsymbol{h}_i\|$ is highly sensitive to various structural factors—such as degree heterogeneity, local noise, and the stochastic nature of message passing—which often causes unpredictable shifts in logit values. As a result, OOD nodes with large embedding norms may be incorrectly assigned low energy scores and misclassified as ID. We refer to this failure mode as *coupling-induced misdetection*. Further our analysis in section 2 reveals that GNNs tend to naturally promote a desirable geometric structure: *angular clustering*, where node embeddings from the same class align closely in direction, while embeddings from different classes diverge. This directional separability implies that angular similarity offers a more stable and semantically meaningful signal for classification than the raw inner product. These insights motivate us to explicitly decouple the norm and angle components in the logit computation. By preserving angular structure while regularizing the influence of magnitude, we aim to produce cleaner energy landscapes and more reliable ID–OOD discrimination.

Motivated by these geometric insights, we propose *GeoEnergy* (**Geo**metric Logit Decoupling for **Energy**-Based OOD Detection), a principled approach that restructures the logit space through hyperspherical projection. By normalizing class weights to enforce hyperspherical geometry while preserving the embedding norm as a confidence signal, GeoEnergy disentangles angular similarity from magnitude. This yields sharper intra-class alignment, larger inter-class angular margins, and more reliable energy landscapes—without altering the GNN backbone or training process. GeoEnergy is fully compatible with existing energy-based frameworks and can be plugged in without architectural changes. In details, our contributions are summarized as follows:

- **Geometric diagnosis.** We revisit the geometric structure of GNN representations and identify two key phenomena—*angular clustering* and *coupling-induced misdetection*—that fundamentally limit the reliability of energy-based OOD detection.
- **Simple yet principled solution.** We propose *GeoEnergy*, a lightweight hyperspherical logit formulation that decouples direction and norm by normalizing class weights while preserving confidence sharpness, yielding structured and well-calibrated energy landscapes.
- **Modular integration.** GeoEnergy can be seamlessly plugged into existing energy-based GNN frameworks without architecture modification.

## 2 Revisiting Geometric Properties of GNNs

The effectiveness of energy-based OOD detection in GNNs is closely tied to the geometry of node embeddings. In this section, we uncover two empirical phenomena that highlight both the benefits and limitations of this geometry. These findings motivate our GeoEnergy approach, which addresses the coupling between embedding norm and angular similarity without modifying the GNN architecture.

**Observation 1: Coupling-Induced Misdetection.** In GNNs, logits are typically computed as:

$$f_j = \boldsymbol{W}_j^\top \boldsymbol{h}_i = \underbrace{\|\boldsymbol{W}_j\|}_{\text{weight norm}} \times \underbrace{\|\boldsymbol{h}_i\|}_{\text{feature norm}} \times \cos(\theta_{j,i}),$$

entangling both the class weight norm and feature norm with the angular similarity. In graph-based settings, these norms are prone to fluctuations due to degree heterogeneity, noisy aggregation, and training dynamics, resulting in unstable logit values. Consequently, energy scores derived from these logits become sensitive to norm variations and less reflective of semantic alignment. Empirically, we observe that under various distribution shifts—structural, feature-level, label, and domain—this instability gives rise to a persistent failure mode: OOD nodes with large norms may yield deceptively low energy scores despite poor directional alignment with class prototypes. As shown in Figure 1(a–d), this manifests as a "low-energy tail" of OOD samples intruding into the ID region, causing systematic false negatives. This coupling-induced misdetection underscores a fundamental weakness of conventional energy-based detection: when norm and direction interact nontrivially, ID–OOD separation breaks down.

**Observation 2: Angular Clustering.** We empirically observe that GNNs promote angular clustering in their node embeddings: same-class nodes exhibit high cosine similarity and align closely in direction, whereas different-class nodes show lower cosine similarity and more dispersed orientations. As shown in Figure 2, across datasets (Cora and Citeseer), the cosine similarity distribution of same-class pairs peaks sharply near 1, while that of different-class pairs is more diffuse. This implies that directional information alone provides a strong semantic signal, making angular similarity a robust and discriminative feature for classification.

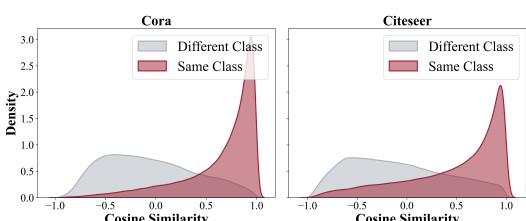

Figure 2: Illustration of Angular Clustering in GNNs.

These observations highlight a geometric tension in GNN embeddings that undermines energy-based OOD detection. To resolve this, we propose GeoEnergy, a geometry-aware formulation that decouples angular similarity from magnitude. By normalizing class weights, GeoEnergy aligns the decision process with the intrinsic angular structure of GNNs, while retaining feature norms to preserve confidence sharpness. This disentangled design yields more stable energy scores and significantly improves ID–OOD separability without modifying the GNN backbone or training strategy.

## 3  Method

Let $\mathcal{G} = (\mathcal{V}, \mathcal{E}, X)$ denote a graph, where $\mathcal{V} = \{v_1, \ldots, v_N\}$ is the node set, $\mathcal{E} \subseteq \mathcal{V} \times \mathcal{V}$ is the edge set, and $X = [\boldsymbol{x}_1, \ldots, \boldsymbol{x}_N] \in \mathbb{R}^{N \times D}$ denotes node features. The graph structure is encoded via an adjacency matrix $A \in \{0, 1\}^{N \times N}$, where $A_{ij} = 1$ indicates an edge between nodes $i$ and $j$. GNNs [1, 2] learn node representations through iterative message passing. At each layer $k$, the hidden representation $\boldsymbol{h}_i^{(k)}$ of node $v_i$ is updated as:

$$\boldsymbol{h}_i^{(k)} = \text{AGGREGATE}^{(k)}\left(\{\boldsymbol{h}_j^{(k-1)} : j \in \mathcal{N}(i)\}\right), \tag{1}$$

where $\mathcal{N}(i)$ is the neighborhood of node $i$ and $\boldsymbol{h}_i^{(0)} = \boldsymbol{x}_i$. The final representation $\boldsymbol{h}_i$ is used for downstream tasks such as node classification. In semi-supervised settings, we are given a labeled subset $\mathcal{I}_s = \{(\boldsymbol{x}_i, y_i)\}_{i=1}^{N_s}$, where $y_i \in \{1, \ldots, C\}$ is the ground-truth label. The GNN is trained to minimize the cross-entropy loss:

$$\mathcal{L}_{\text{CE}} = -\frac{1}{|\mathcal{I}_s|} \sum_{i \in \mathcal{I}_s} \log \frac{e^{f_{y_i}(\boldsymbol{x}_i, A)}}{\sum_{c=1}^{C} e^{f_c(\boldsymbol{x}_i, A)}}, \tag{2}$$

where $f_c(\boldsymbol{x}_i, A)$ denotes the logit of class $c$. However, in real-world applications, the distribution between $\mathcal{I}_s$ and $\mathcal{I}_u$ often varies, leading to a significant challenge for the model: identifying and addressing instances where the distribution deviates, known as OOD cases. This challenge gives rise

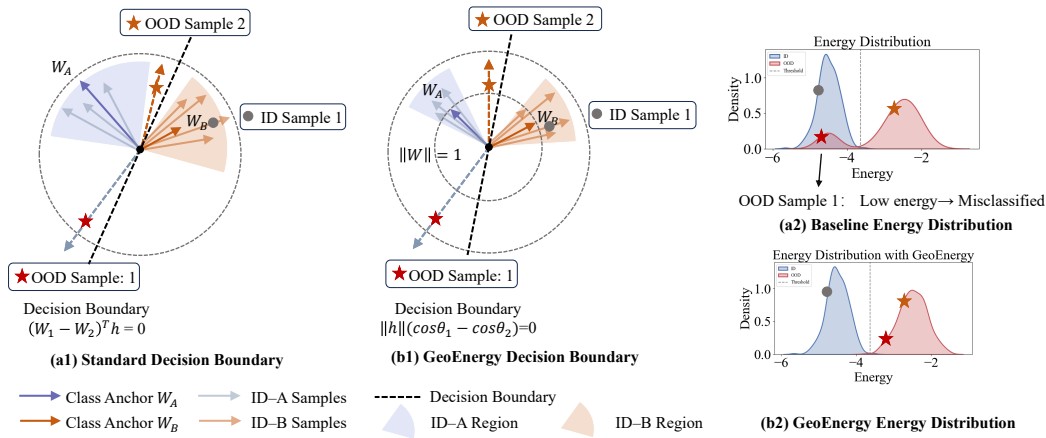

Figure 3: Decision Boundary Evolution: From Euclidean Softmax to GeoEnergy.

to the problem of OOD detection, aiming to recognize unlabeled nodes that do not conform to the feature distribution of the labeled node set $\mathcal{I}_s$. The OOD detection task can be formulated as a binary classification problem to detect whether an input node $\mathbf{x}$ is an OOD case:

$$g\left(\mathbf{x};\mathcal{G}_\mathbf{x},f_\theta\right) = \begin{cases} 1, & \mathbf{x} \text{ is an in-distribution instance,} \\ 0, & \mathbf{x} \text{ is an out-of-distribution instance.} \end{cases} \tag{3}$$

where $\mathcal{G}_{\boldsymbol{x}}$ denotes the $L$-hop ego-graph around $\boldsymbol{x}$, and $g$ is a scoring function designed to reflect uncertainty. This formulation considers the inter-dependence among nodes in the graph, which is a unique aspect of OOD detection in graph-based contexts compared to traditional OOD detection.

### 3.1 Geometric Decoupling of Logits

We decouple magnitude and direction by projecting each class weight onto a unit hypersphere. Logits are reformulated as:

$$f_j = s \cdot \|\boldsymbol{h}_i\| \cdot \cos(\theta_{j,i}), \tag{4}$$

where $\theta_{j,i}$ is the angle between embedding $\boldsymbol{h}_i$ and weight $\boldsymbol{W}_j$, and $s$ is a scaling factor. This design enforces angular structure via normalized weights while retaining $\|\boldsymbol{h}_i\|$ to preserve confidence expressiveness. Compared to traditional formulations where $f_j = \boldsymbol{W}_j^\top \boldsymbol{h}_i = \|\boldsymbol{W}_j\| \cdot \|\boldsymbol{h}_i\| \cdot \cos(\theta_{j,i})$, GeoEnergy enforces $\|\boldsymbol{W}_j\| = 1$ to eliminate magnitude-based distortions. This enhances geometric interpretability, aligning with the goal of making class separation rely solely on direction. Such a formulation encourages tighter intra-class alignment and enlarged inter-class angular margins, which is theoretically beneficial for classification in high-dimensional spaces.

**Proposition 1 (Reduced Intra-Class Variation).** By normalizing the class weight vectors while preserving the feature norm for energy scaling, GeoEnergy promotes directional alignment among features of the same class, leading to improved angular compactness:

$$\cos(\theta_{j,i}) = \frac{\boldsymbol{W}_j^\top \boldsymbol{h}_i}{\|\boldsymbol{W}_j\| \cdot \|\boldsymbol{h}_i\|}, \quad \text{with } \|\boldsymbol{W}_j\| = 1. \tag{5}$$

The expected pairwise cosine similarity for same-class features increases:

$$\mathbb{E}_{x_i,x_j \sim y_i} \left[ 1 - \cos\left( \angle(\tilde{h}_i^{(k)}, \tilde{h}_j^{(k)}) \right) \right] \to 0 \tag{6}$$

indicating tighter angular compactness.

**Proposition 2 (Increased Inter-Class Separation).** The scaling factor $s$ amplifies angular differences in the logit space, thus increasing margin:

$$z_i = s \cdot \cos(\theta_i), \quad z_j = s \cdot \cos(\theta_j), \quad \|z_i - z_j\| = s \cdot |\cos(\theta_i) - \cos(\theta_j)|. \tag{7}$$

If the feature mapping $\phi$ satisfies Lipschitz continuity, i.e.,

$$\|\phi(x_i) - \phi(x_j)\| \leq L\|x_i - x_j\|, \tag{8}$$

Table 1: Performance of OOD detection using metrics such as AUROC(↑), AUPR(↑), and FPR95(↓), alongside ID ACC on the Cora dataset across three types of OOD scenarios. The best results are highlighted in bold.

| Model | OOD Expo | Cora-Structure | | | | Cora-Feature | | | | Cora-Label | | | |
|---|---|---|---|---|---|---|---|---|---|---|---|---|---|
| | | AUROC | AUPR | FPR95 | ID ACC | AUROC | AUPR | FPR95 | ID ACC | AUROC | AUPR | FPR95 | ID ACC |
| MSP | No | 70.90 | 45.73 | 87.30 | 75.50 | 85.39 | 73.70 | 64.88 | 75.30 | 91.36 | 78.03 | 34.99 | 88.92 |
| ODIN | No | 49.92 | 27.01 | 100.0 | 74.90 | 49.88 | 26.96 | 100.0 | 75.00 | 49.80 | 24.27 | 100.0 | 88.92 |
| Mahalanobis | No | 46.68 | 29.03 | 98.19 | 74.90 | 49.93 | 31.95 | 99.93 | 74.90 | 67.62 | 42.31 | 90.77 | 88.92 |
| Energy | No | 71.73 | 46.08 | 88.74 | 76.00 | 86.15 | 74.42 | 65.81 | 76.10 | 91.40 | 78.14 | 41.08 | 88.92 |
| GKDE | No | 68.61 | 44.26 | 84.34 | 73.70 | 82.79 | 66.52 | 68.24 | 74.80 | 57.23 | 27.50 | 88.95 | 89.87 |
| GPN | No | 77.47 | 53.26 | 76.22 | 76.50 | 85.88 | 73.79 | 56.17 | 77.00 | 90.34 | 77.40 | 37.42 | 91.46 |
| GNNSafe | No | 87.52 | 77.46 | 73.15 | 75.80 | 93.44 | 88.19 | 38.92 | 76.40 | 92.80 | 82.21 | 30.83 | 88.92 |
| + GeoEnergy | No | **90.83** | **80.28** | **47.53** | 79.20 | **94.36** | **88.90** | **28.91** | 78.10 | **95.58** | **90.87** | **19.68** | 91.46 |
| NodeSafe | No | 94.07 | 83.98 | 25.63 | 77.20 | 95.30 | 88.82 | 23.08 | 78.70 | 93.80 | 85.22 | 29.41 | 89.87 |
| + GeoEnergy | No | **95.21** | **85.32** | **22.46** | 77.30 | **96.76** | **88.90** | **21.54** | 78.21 | **96.11** | **86.46** | **23.77** | 90.21 |
| OE | Yes | 67.98 | 46.93 | 95.31 | 71.80 | 81.83 | 70.84 | 83.79 | 73.30 | 89.47 | 77.01 | 46.55 | 87.97 |
| Energy FT | Yes | 75.88 | 49.18 | 67.73 | 75.50 | 88.15 | 75.99 | 47.53 | 75.30 | 91.36 | 78.49 | 37.83 | 90.51 |
| GNNSafe++ | Yes | 90.62 | **81.88** | 53.51 | 76.10 | 95.56 | 90.27 | 27.73 | 76.80 | 92.75 | 82.64 | 34.08 | 91.46 |
| + GeoEnergy | Yes | **91.27** | 81.04 | **42.61** | 76.50 | **96.48** | **90.62** | **16.25** | 77.10 | **92.85** | **83.39** | **31.74** | 91.46 |
| NodeSafe++ | Yes | 94.64 | 85.63 | 23.34 | 76.40 | 96.56 | 91.96 | 14.73 | 77.10 | 94.88 | 86.66 | 22.52 | 91.46 |
| + GeoEnergy | Yes | **95.22** | **86.15** | **21.89** | 77.20 | **97.36** | **92.45** | **10.41** | 77.67 | **95.36** | **89.25** | **17.85** | 91.53 |

then logit differences satisfy:
$$\|z_i - z_j\| \geq s \cdot L \cdot \|x_i - x_j\|,  \tag{9}$$
providing a lower bound for decision margin controlled by $s$.

## 3.2 Energy-Based OOD Detection via Angular Regularization

Following GNNSafe [13], we define the Helmholtz free energy for OOD detection as:

$$E(\boldsymbol{x}, \mathcal{G}_{\boldsymbol{x}}) = -\log \sum_{c=1}^{C} \exp(f_c).  \tag{10}$$

Under our angular logit formulation, the energy score becomes:

$$E_{\text{angular}}(\boldsymbol{x}_i) = -\log \sum_{j=1}^{C} \exp(s \cdot \|\boldsymbol{h}_i\| \cdot \cos(\theta_{j,i})).  \tag{11}$$

This reshaped energy surface allows ID samples to cluster in low-energy regions while pushing OOD nodes to high-energy areas. Moreover, the preservation of $\|\boldsymbol{h}_i\|$ enables confidence sharpness that supports calibration. GeoEnergy acts as a plug-in logit head and energy scoring mechanism, which can be seamlessly applied to energy-based graph models such as GNNSafe [13], NodeSafe [15], and TopoOOD [14]. During training, it replaces the standard dot-product logit formulation, while during inference, its angular energy serves as the OOD scoring function. Crucially, GeoEnergy requires no modification of the graph encoder, making it a practical and generalizable to existing frameworks.

## 4 Experiments

In previous sections, we introduced GeoEnergy, a hyperspherical logit reparameterization strategy designed to improve OOD detection and confidence calibration in GNNs. This section presents a comprehensive evaluation of GeoEnergy, examining its robustness under various OOD shifts and its effect on uncertainty estimation. To rigorously assess its capabilities, we explore the following Research Questions (RQs):

RQ1: How effectively does GeoEnergy improve the detection of OOD samples, compared to traditional methods?

RQ2: What impact does GeoEnergy have on confidence calibration for GNNs?

RQ3: Does the enhanced confidence calibration and optimized logit distribution provided by GeoEnergy lead to better self-training outcomes?

To address these questions, we conduct experiments on OOD detection and confidence calibration to evaluate GeoEnergy's ability to distinguish OOD from ID samples without compromising ID performance. We also compare its calibration against existing methods and assess how improved calibration enhances self-training, demonstrating GeoEnergy's robustness and reliability.

Table 2: Performance of OOD detection across three OOD scenarios, using AUROC (↑). Additional metrics such as AUPR, FPR95, and ID accuracy are detailed in Appendix B.1. The best results are highlighted in bold.

| Model | OOD Expo | Citeseer | | | Pubmed | | |
| --- | --- | --- | --- | --- | --- | --- | --- |
| | | Structure | Feature | Label | Structure | Feature | Label |
| MSP | No | 66.34 | 78.32 | 88.42 | 74.31 | 83.28 | 85.71 |
| ODIN | No | 49.23 | 49.86 | 51.33 | 49.76 | 49.67 | 56.24 |
| Mahalanobis | No | 45.26 | 49.92 | 53.46 | 55.28 | 69.12 | 75.77 |
| Energy | No | 65.62 | 79.19 | 89.98 | 74.33 | 84.16 | 86.81 |
| GKDE | No | 61.48 | 74.68 | 82.69 | 74.02 | 82.25 | 83.36 |
| GPN | No | 70.55 | 78.46 | 85.65 | 74.96 | 82.56 | 86.51 |
| GNNSafe | No | 79.79 | 83.46 | 90.01 | 87.52 | 94.28 | 88.02 |
| + GeoEnergy | No | **81.23** | **88.94** | **92.57** | **88.92** | **95.83** | **89.90** |
| NodeSafe | No | 88.40 | 90.41 | 91.66 | 94.13 | 95.97 | 93.80 |
| + GeoEnergy | No | **89.75** | **92.12** | **93.18** | **95.45** | **96.13** | **94.19** |
| OE | Yes | 58.74 | 72.06 | 89.44 | 74.41 | 82.34 | 81.97 |
| Energy FT | Yes | 68.87 | 79.23 | 91.34 | 73.54 | 78.95 | 91.83 |
| GNNsafe++ | Yes | 82.43 | 83.27 | 91.57 | 90.62 | 95.16 | 87.98 |
| + GeoEnergy | Yes | **84.96** | **86.80** | **92.19** | **91.99** | **96.97** | **88.18** |
| Nodesafe++ | Yes | 86.90 | 91.14 | 91.98 | 96.30 | 95.26 | 93.48 |
| + GeoEnergy | Yes | **87.67** | **93.13** | **93.23** | **97.87** | **96.51** | **94.38** |

## 4.1 Out-of-distribution Detection

In response to **RQ1**, we conduct graph OOD detection experiments under two settings to thoroughly evaluate our approach. **Without OOD Exposure:** The model is trained without any OOD data, simulating real-world scenarios where unexpected data types emerge post-deployment. **With OOD exposure:** OOD samples are introduced during training, allowing the model to learn and adapt to anomalies, assessing its ability to generalize and detect OOD instances with prior knowledge.

### 4.1.1 Datasets and Splits

Following recent work on graph OOD detection [13, 16, 15], our experiments use five benchmark datasets to reflect real-world scenarios with OOD instances. These datasets cover two scenarios:

**Single-graph scenario:** OOD instances exist within the same graph as training data but remain unseen during training. (1) *Cora, Citeseer, Pubmed* [1]: OOD data is synthetically generated using structure manipulation, feature interpolation, and label leave-out. ID data is split into training, validation, and testing sets in a 1:1:8 ratio. (2) *ogbn-Arxiv* [17]: This large citation dataset spans 1960–2020. Papers published before 2015 serve as ID data, while those after 2017 are OOD. Papers from 2015–2016 are used for OOD exposure during training. ID data follows a 1:1:8 split.

**Multi-graph scenario:** OOD instances come from entirely separate graphs or subgraphs with no direct connections to the training set. *Twitch-Explicit* [18]: It derived from the Twitch streaming platform, represents users as nodes, with edges indicating mutual friendships. Node features capture user activities and interactions. It consists of multiple subgraphs; we use the DE subgraph as ID data, while EN, ES, FR, and RU serve as OOD data. The ENGB subgraph is used for OOD exposure during training. ID data is randomly split into training, validation, and testing sets in a 1:1:8 ratio.

### 4.1.2 Evaluation Metrics

We assess OOD detection and ID accuracy using key metrics: **ID Accuracy** evaluates performance on known data, **AUROC** measures the ability to separate ID from OOD instances, **AUPR** captures the precision-recall trade-off for imbalanced data, and **FPR95** quantifies the false positive rate at a 95% true positive rate. Detailed metric descriptions are in Appendix A.2.

### 4.1.3 Comparison Methods

We evaluate GeoEnergy's effectiveness in OOD detection under two settings. **Without OOD Exposure.** We compare GeoEnergy-enhanced models against two categories of baselines that do not use auxiliary OOD supervision during training. The first category includes classical OOD detection methods originally developed for vision tasks under the i.i.d. assumption, such as MSP [19], ODIN [20], Mahalanobis [21], OE [22], and Energy [23]. To adapt these methods to graph-structured data, we replace their CNN backbones with GCN encoders. The second category consists of graph-specific OOD detection methods that explicitly model topological dependencies, including GKDE [24], GPN [12],

Table 3: OOD detection results and ID accuracy on Twitch, where nodes from different subgraphs serve as OOD data, and Arxiv, where papers published after 2017 are considered OOD. Detailed results for each OOD dataset (i.e., subgraph or year) are provided in Appendix B.1.

| Model | OOD Expo | Twitch | | | | Arxiv | | | |
|---|---|---|---|---|---|---|---|---|---|
| | | AUROC | AUPR | FPR95 | ACC | AUROC | AUPR | FPR95 | ACC |
| MSP | No | 33.59 | 49.14 | 97.45 | 68.72 | 63.91 | 75.85 | 90.59 | **53.78** |
| ODIN | No | 58.16 | 72.12 | 93.96 | 70.79 | 55.07 | 68.85 | 100.0 | 51.39 |
| Mahalanobis | No | 55.68 | 66.42 | 90.13 | 70.51 | 56.92 | 69.63 | 94.24 | 51.59 |
| Energy | No | 51.24 | 60.81 | 91.61 | 70.40 | 64.20 | 75.78 | 90.80 | 53.36 |
| GKDE | No | 46.48 | 62.11 | 95.62 | 67.44 | 58.32 | 72.62 | 93.84 | 50.76 |
| GPN | No | 51.73 | 66.36 | 95.51 | 68.09 | - | - | - | - |
| GNNSafe | No | 66.82 | 70.97 | 76.24 | 70.40 | 71.06 | 80.44 | 87.01 | 53.39 |
| + GeoEnergy | No | **72.28** | **78.13** | **66.62** | 72.80 | **71.99** | **81.05** | **84.85** | 52.76 |
| NodeSafe | No | 89.99 | 93.33 | 47.00 | 71.79 | 72.44 | 81.51 | 84.27 | 51.20 |
| + GeoEnergy | No | **90.12** | **94.47** | **44.56** | 71.84 | **73.19** | **82.32** | **81.65** | 51.76 |
| OE | Yes | 55.72 | 70.18 | 95.07 | 70.73 | 69.80 | 80.15 | 85.16 | 52.39 |
| Energy FT | Yes | 84.50 | 88.04 | 61.29 | 70.52 | 71.56 | 80.47 | 80.59 | 53.26 |
| GNNSafe++ | Yes | 95.36 | 97.12 | 33.57 | 70.18 | 74.77 | 83.21 | 77.43 | 53.50 |
| + GeoEnergy | Yes | **95.50** | **97.18** | **32.51** | 70.16 | **75.73** | **83.70** | **73.79** | 52.12 |
| NodeSafe++ | Yes | 98.50 | 99.18 | 3.43 | 71.85 | 75.49 | 83.71 | 75.24 | 52.93 |
| + GeoEnergy | No | **98.87** | **99.24** | **3.11** | 71.95 | **77.21** | **84.21** | **72.18** | 52.76 |

GNNSafe [25], and the recently proposed NodeSafe [15]. We integrate GeoEnergy into GNNSafe and NodeSafe in a plug-and-play fashion—without any retraining or architectural changes—and compare their performance with and without GeoEnergy. Across all benchmarks, GeoEnergy consistently improves OOD detection performance by mitigating norm-induced logit distortions while preserving angular separability. **With OOD Exposure.** We further assess GeoEnergy under settings where a small set of auxiliary OOD samples is available during training. This setting follows protocols introduced in prior work such as OE [22], Energy Fine-Tuning [23], and GNNSafe++ [25]. We evaluate GeoEnergy-enhanced variants, including GNNSafe++(w GeoEnergy) and NodeSafe++(w GeoEnergy), and compare them against their corresponding base models.

### 4.1.4 Comparative Results

We explore the enhancements brought by GeoEnergy to OOD detection through detailed experimental evaluations. In Table 1, 2 and 3, we report the OOD detection performance and ID accuracy of GeoEnergy with other competitive methods. Table 1 presents a comparative analysis of OOD detection performance of various methods on Cora under three different OOD perturbations, including structure manipulation, feature interpolation and label leave-out, along with their ability to maintain accuracy on ID data. Due to space constraints, Table 2 distills the AUROC comparisons for each method across Cora, Amazon and Coauthor under different OOD perturbations. The detailed comparisons of AUPR, FPR95, and ID accuracy on these datasets are provided in the Appendix B.1. For dataset Twitch and Arxiv, Table 3 provides a comparison of OOD detection performance between our method and various baselines. Additionally, detailed results on Twitch's sub-graphs and Arxiv papers published across different years are presented in Appendix B.1.

**GeoEnergy significantly enhances OOD detection performance.** We highlight several key findings: (1) *Overall Superior Performance:* Under settings with and without OOD exposure, the rows for GeoEnergy and +GeoEnergy in the tables represent the performance of our method. GeoEnergy consistently shows superior performance across all three perturbation types on the Cora dataset, surpassing other methods. Additionally, GeoEnergy achieves higher AUROC scores across Cora, Amazon, and Coauthor datasets, and also exhibits improved performance on the Twitch and Arxiv datasets compared to SOTA methods. These results highlight the broad effectiveness of GeoEnergy in OOD detection, demonstrating its efficacy across both single-graph and multi-graph scenarios. (2) *Significant Reduction in FPR95:* GeoEnergy exhibits remarkable improvements in the FPR95 metric. Specifically, it reduces FPR95 by 35% in the structure manipulation scenario, 25.7% in the feature interpolation scenario, and 36.1% in the label leave-out scenario compared to the previous state-of-the-art method GNNSafe. This highlights its significant advantage in minimizing false positive rates. Additionally, we detail the optimal hyperparameter selection for the scaling factor $s$ across various datasets and different OOD scenarios in Appendix B.1.

**GeoEnergy maintains classification accuracy.** GeoEnergy preserves competitive ID accuracy and even achieves slight improvements in some cases. As shown in Table 1 and 3, our method maintains

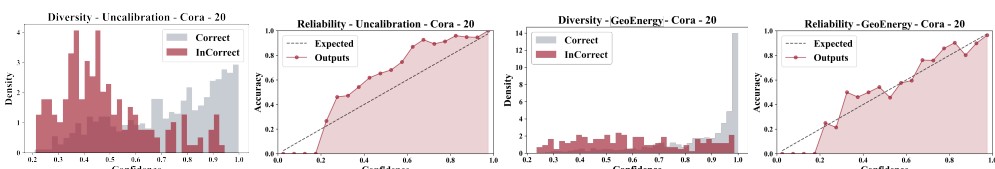

Figure 4: Analysis of confidence distributions and reliability diagrams for GCN on Cora with $L/C = 20$.

ID accuracy and, in some instances, enhances in-distribution classification precision. This highlights GeoEnergy's ability to improve the representation of graph data.

## 4.2 Confidence Calibration and Self-training

Recent studies have demonstrated that GNNs often exhibit an underconfidence tendency [9, 10], with predictions systematically less confident than actual probabilities. This underconfidence, due to the complex and noisy nature of graph-structured data and challenges in aggregating neighbor information [26], impedes accurate OOD detection by blurring the distinction between ID and OOD data. In response to **RQ2** and **RQ3**, we empirically validated the impact of GeoEnergy on improving confidence calibration in GNNs. Our experiments focused on how this adjustment aligns confidence scores more closely with true probabilities. Furthermore, we explored self-training to illustrate the practical benefits of enhanced calibration.

Table 4: ECE (M=20) of different calibration methods on GCN, GAT, and GraphSAGE. Bold texts indicate the best results.

| Dataset | Model | Calibration Methods | | | | | |
| --- | --- | --- | --- | --- | --- | --- | --- |
| | | Uncal. | TS | MS | CaGCN | GCL | GeoEnergy |
| Cora | GCN | 0.1347 | 0.0488 | 0.0414 | 0.0401 | 0.0394 | **0.0361** |
| | GAT | 0.1558 | 0.0717 | 0.0544 | 0.0450 | 0.0444 | **0.0430** |
| | GraphSage | 0.1037 | 0.0463 | 0.0371 | 0.0398 | - | **0.0360** |
| Citeseer | GCN | 0.1248 | 0.0641 | 0.0644 | 0.0595 | 0.0579 | **0.0539** |
| | GAT | 0.1534 | 0.0916 | 0.0633 | 0.0572 | 0.0660 | **0.0552** |
| | GraphSage | 0.1135 | 0.0808 | 0.0866 | 0.0691 | - | **0.0579** |
| Pubmed | GCN | 0.0586 | 0.0541 | 0.0476 | 0.0405 | 0.0394 | **0.0332** |
| | GAT | 0.0835 | 0.0656 | 0.0501 | 0.0356 | 0.0417 | **0.0316** |
| | GraphSage | 0.0338 | 0.0337 | 0.0342 | 0.0364 | - | **0.0259** |

focused on how this adjustment aligns confidence scores more closely with true probabilities. Furthermore, we explored self-training to illustrate the practical benefits of enhanced calibration.

**Datasets and Benchmarks.** We use three citation network datasets: Cora, Citeseer, and Pubmed [27]. For semi-supervised node classification, we set the label rate $L/C \in \{20, 40, 60\}$. We select three GNN models for node classification: GCNs [1], GATs [2], and GraphSAGE [28]. This diversity allows for a comprehensive evaluation of confidence calibration across different GNN architectures. We selected four baseline methods for comparison, including two traditional calibration methods, Temperature Scaling (TS) and Matrix Scaling (MS) [29], as well as two GNN-specific calibration methods, CaGCN [9] and GCL [10].

**Evaluation Metrics.** We evaluate confidence calibration using Expected Calibration Error (ECE) [30], which quantifies the gap between model confidence and accuracy. Specifically, predictions are partitioned into $M$ bins, and ECE is computed as the weighted average of the absolute difference between average confidence and accuracy in each bin: $ECE = \sum_{m=1}^{M} \frac{|B_m|}{n} |\text{acc}(B_m) - \text{conf}(B_m)|$. Here, $B_m$ denotes the set of predictions in the $m$-th bin, and $n$ is the total number of samples. A lower ECE indicates better calibration. We use $M = 20$ in all experiments.

**Visualization Analysis.** To illustrate GeoEnergy's impact on confidence calibration, Figure 1 analyzes a GCN's performance on the Cora dataset with $L/C = 20$. The left histograms reveal a notable shift in confidence distribution for correct (gray) and incorrect (red) predictions. Initially, correct predictions are dispersed across confidence levels, indicating uncertainty. After applying GeoEnergy, confidence for correct predictions becomes more concentrated at higher values, reflecting improved certainty and reliability. The right reliability diagrams further quantify calibration improvements. Initially, the model shows underconfidence, with accuracy exceeding predicted confidence (solid line above the ideal dashed line). After applying GeoEnergy, the solid line aligns closely with the ideal, indicating better-calibrated predictions.

**Comparative Results** Detailed experimental results on calibration performance at a label rate of $L/C = 20$ are meticulously documented in Table 4, which provides the ECE for GCN, GAT, and GraphSage models across various datasets, utilizing different calibration techniques, where "Uncal." denotes an uncalibrated model. Detailed analyses for $L/C = 40$ and 60 are discussed in

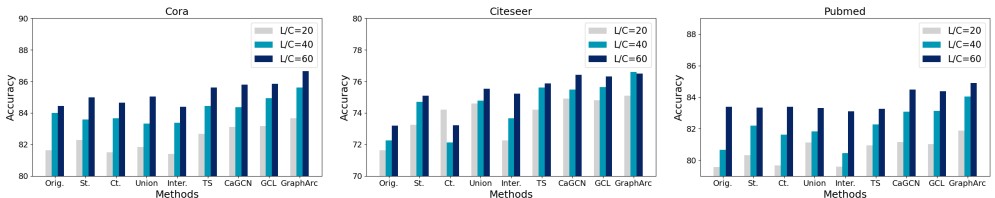

Figure 5: Self-training accuracy comparison of various methods under different labelrates.

Appendix B.2, which also summarizes the optimal hyperparameter selection for the scaling factor $s$ in GeoEnergy across multiple datasets.

*GeoEnergy enhances confidence calibration performance.* Detailed analysis of Table 4 illustrates that GeoEnergy significantly enhances calibration accuracy, outperforming both traditional and GNN-specific methods across various scenarios. This superior calibration performance not only ensures more accurate confidence estimates but also crucially supports the generation of reliable pseudo labels, enhancing the model's effectiveness in self-training contexts. This robust calibration capability is instrumental in maintaining high accuracy on ID data while effectively identifying and handling out-of-distribution samples.

*GeoEnergy boosts self-training efficacy.* GeoEnergy validates its impact on self-training by examining its influence on classification accuracy within these settings. As depicted in Figure 5, GeoEnergy substantially elevates classification accuracy across diverse datasets and label rates, illustrating its dual benefits: precision in calibration and enhancement of self-training outcomes through the utilization of precise pseudo labels. These experiments affirm that GeoEnergy synchronizes GNNs' confidence estimates with actual probabilities, markedly delineating ID from OOD samples.

## 5 Related Work

**OOD detection for GNNs.** To tackle the complexities of graph data with inter-dependent nodes, OOD detection for GNNs is a burgeoning field. GraphDE [11] utilizes a variational approach combined with mixed generative models to identify distribution shifts and effectively down-weight outliers, enhancing OOD detection capabilities for new datasets. In node classification, techniques like Graph-based Kernel Dirichlet Distribution Estimation (GKDE) [24] and Graph Posterior Network (GPN) [12] employ Bayesian GNN models that effectively consider the inter-dependence among nodes. Additionally, GNNSafe [13] introduces an energy-based OOD discriminator that operates independently of specific GNN architectures, offering a versatile solution to OOD challenges in graph neural networks. Uniquely, our method GeoEnergy harnesses the crucial insights provided by the intrinsic properties of node embeddings, specifically the natural clustering within the feature space based on angular relationships. This method extends the softmax loss to angular similarity loss and constrains weight vectors to a hypersphere, optimizing the angles between features. By leveraging the inherent low-dimensional manifold structure of graph data, GeoEnergy enhances the discriminative power of GNNs for effective OOD detection. A comprehensive review of related work is provided in Appendix C.

## 6 Conclusion

This paper presents GeoEnergy, a simple yet effective approach for improving energy-based OOD detection in graph neural networks. Through an in-depth analysis of GNN logit geometry, we identify two critical phenomena—angular clustering and coupling-induced misdetection—that undermine the reliability of conventional energy scoring. GeoEnergy addresses these issues by decoupling magnitude and direction via weight normalization, thereby enhancing intra-class compactness, inter-class separation, and energy stability. Importantly, GeoEnergy is model-agnostic and can be seamlessly integrated into existing energy-based frameworks without retraining. Extensive experiments across diverse OOD benchmarks demonstrate that GeoEnergy consistently improves detection performance and calibration, providing a strong geometric foundation for robust and trustworthy GNN deployment in open-world environments.

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

# A Details of Experiments

## A.1 Details of datasets

Following the setup from recent work on graph OOD detection and generalization, our experiments utilize five widely recognized node classification benchmark datasets to reflect real-world scenarios involving OOD instances within graphs. These datasets encompass two distinct scenarios:

**Single-graph scenario:** In this scenario, we use the datasets Cora[27], Amazon-Photo[31], Coauthor-CS[32], and ogbn-Arxiv[17]. Here, OOD testing instances exist within the same graph as the training instances but remain unseen during training.

- Cora, Amazon-Photo, Coauthor-CS: Cora is a citation network where nodes represent scientific papers and edges indicate citations between them. Amazon-Photo is a co-purchase network with nodes representing products and edges signifying frequent co-purchases. Coauthor-CS is a co-authorship network where nodes represent authors and edges denote co-authorship of papers in the field of Computer Science. Due to the lack of explicit domain distinctions, we synthetically generate OOD data using three different approaches: *Structure manipulation*: The original graph is utilized as in-distribution data, while a new graph generated using the stochastic block model serves as OOD data. *Feature interpolation*: Random interpolation is applied to create node features for OOD data, with the original graph as in-distribution data. *Label leave-out*: Nodes with specific class labels are used as in-distribution data, and nodes with other class labels are treated as OOD data. For Amazon-Photo and Coauth-CS, we randomly split the ID data into training, validation, and testing sets in a 1:1:8 ratio. For Cora, we split the ID data following the semi-supervised learning setting by [1].

- ogbn-Arxiv: A large graph dataset that records citation information from 1960 to 2020. Each node represents a paper, labeled with its subject area, and edges denote citation relationships. Nodes have 128-dimensional feature vectors obtained from the word embeddings of their titles and abstracts. ID data is split in a 1:1:8 ratio based on publication year.

**Multi-graph scenario:** This scenario is represented by the dataset Twitch-Explicit[18], where OOD instances originate from a completely different graph or subgraph that has no connections with nodes in the training set.

- Twitch-Explicit: Derived from the Twitch streaming platform, each node represents a user, edges denote mutual friendships between users, and node features capture user activities and interaction patterns. The dataset includes multiple subgraphs; we use the DE subgraph as the ID data, while the other subgraphs (EN, ES, FR, and RU) serve as OOD data. Additionally, the ENGB subgraph is used for OOD exposure during training. We randomly split the ID data into training, validation, and testing sets in a 1:1:8 ratio.

Table 5: Dataset Details for In-Distribution (ID), Out-of-Distribution (OOD), and OOD Exposure Data

| Dataset | ID Data | OOD Data | OOD Exposure Data |
|---|---|---|---|
| Cora
Amazon-Photo
Coauthor-CS | Original graph | Structure manipulation,
Feature interpolation,
Label leave-out | Synthetically generated
OOD instances |
| Twitch | DE subgraph | Five other subgraphs | ENGB subgraph |
| Arxiv | Papers before 2015 | Papers after 2017 | Papers in 2015 and 2016 |

## A.2 Details of evaluation metrics

In our experiments, we use several evaluation metrics to comprehensively evaluate the model's ability to recognize known data and detect OOD instances, ensuring robustness and reliability in diverse scenarios.

**ID Accuracy** is the ratio of correctly predicted instances to the total instances within the in-distribution (ID) data. This metric measures how well the model performs on data it has seen during training or

data from the same distribution. High ID Accuracy indicates that the model effectively learns the patterns in the ID data.

**AUROC (Area Under the Receiver Operating Characteristic Curve)** measures the model's ability to distinguish between ID and OOD instances across all possible classification thresholds. This metric evaluates the trade-off between the true positive rate (sensitivity) and the false positive rate (1-specificity). It provides a single scalar value representing the model's overall ability to separate ID from OOD instances.

**AUPR (Area Under the Precision-Recall Curve)** evaluates the trade-off between precision (the fraction of true positive instances among all instances predicted as positive) and recall (the fraction of true positive instances among all actual positive instances) for different thresholds. This metric is particularly useful in scenarios with imbalanced data, where the number of OOD instances may be much smaller than ID instances. It provides a more detailed picture of the model's performance in detecting OOD instances by focusing on the precision-recall trade-off.

**FPR95 (False Positive Rate at 95% True Positive Rate)** measures the false positive rate when the true positive rate is at 95%. This metric assesses the model's reliability in identifying OOD instances by ensuring that a high true positive rate does not come at the cost of a high false positive rate. It provides insight into the model's performance in critical regions of the ROC curve.

## A.3 Details of baselines

**MSP (Maximum Softmax Probability) [19]:** MSP is a simple yet effective baseline for OOD detection. It uses the maximum predicted probability from the softmax output as a confidence score. Instances with low confidence scores are considered OOD. This approach is straightforward but can struggle in scenarios with complex data distributions.

**ODIN (Out-of-DIstribution detector for Neural networks) [20]:** ODIN improves upon MSP by applying temperature scaling and small perturbations to the input. These adjustments enhance the separability between ID and OOD data by making the softmax scores more discriminative. ODIN requires fine-tuning on a validation set to determine optimal parameters for temperature scaling and perturbation magnitude.

**Mahalanobis [21]:** This method estimates the class-conditional distributions of the model's features using a multivariate Gaussian distribution. The Mahalanobis distance between a test sample and the closest class mean is used as a confidence score. Lower distances indicate higher confidence of being ID, making this approach effective for OOD detection, especially in scenarios with well-separated feature distributions.

**OE (Outlier Exposure) [22]:** OE leverages an auxiliary dataset containing OOD examples during training. The model is trained to assign low confidence scores to these OOD samples while maintaining high confidence for ID samples. This exposure to OOD data during training helps the model learn better decision boundaries, improving its OOD detection capabilities.

**Energy and Energy FineTune [23]:** The energy-based model uses the concept of energy from statistical physics to measure prediction confidence, with lower energy indicating higher confidence. Energy scores are computed from the model's logits, providing a unified approach for ID and OOD detection. Energy FineTune extends this by incorporating OOD data during training, fine-tuning the model to better discriminate between ID and OOD samples, thus improving calibration and detection accuracy.

**GKDE (Graph-based Kernel Dirichlet distribution Estimation) [24]:** GKDE is specifically designed for graph data, leveraging kernel density estimation to model the uncertainty in node embeddings. By estimating the Dirichlet distribution over the graph's features, GKDE provides a probabilistic framework for detecting OOD nodes. This approach effectively captures the dependencies and topological structures unique to graph data.

**GPN (Graph Posterior Network) [12]:** GPN integrates Bayesian principles with graph neural networks to model posterior distributions over node labels. This method uses a probabilistic encoder to generate latent variables and a decoder to predict labels, capturing the uncertainty in predictions. GPN's Bayesian framework makes it robust for OOD detection by effectively quantifying prediction uncertainty.

Table 6: Performance of out-of-distribution detection using metrics such as AUROC(↑), AUPR(↑), and FPR95(↓), alongside in-distribution accuracy (ID ACC) on the Amazon dataset across three types of OOD scenarios—structure manipulation, feature interpolation, and label leave-out—is presented. The best results are highlighted in bold.

| Model | OOD Expo | Amazon-Structure | | | | Amazon-Feature | | | | Amazon-Label | | | |
|---|---|---|---|---|---|---|---|---|---|---|---|---|---|
| | | AUROC | AUPR | FPR95 | ID ACC | AUROC | AUPR | FPR95 | ID ACC | AUROC | AUPR | FPR95 | ID ACC |
| MSP | No | 98.27 | 98.54 | 6.13 | 92.84 | 97.31 | 95.16 | 8.72 | 92.89 | 93.97 | 91.32 | 26.65 | 95.76 |
| ODIN | No | 93.24 | 95.26 | 65.44 | 92.84 | 81.15 | 78.47 | 100.0 | 92.71 | 65.97 | 57.80 | 90.23 | **96.08** |
| Mahalanobis | No | 71.69 | 79.01 | 99.91 | 92.79 | 76.50 | 71.14 | 76.12 | 92.86 | 73.25 | 66.89 | 74.30 | 95.76 |
| Energy | No | 98.51 | 98.72 | 4.97 | **92.86** | 97.87 | 95.64 | 6.00 | **92.96** | 93.81 | 91.13 | 28.48 | 95.72 |
| GKDE | No | 76.39 | 81.58 | 99.25 | 87.57 | 58.96 | 66.76 | 99.28 | 86.18 | 65.58 | 65.20 | 96.87 | 89.37 |
| GPN | No | 97.17 | 96.39 | 11.65 | 88.51 | 87.91 | 84.77 | 49.11 | 90.05 | 92.72 | 90.34 | 37.16 | 90.07 |
| GNNSafe | No | 99.58 | **99.76** | 0.00 | 92.53 | 98.55 | 98.99 | **0.31** | 92.81 | 97.35 | 97.12 | **6.59** | 95.76 |
| +GeoEnergy | No | **99.63** | 99.56 | **0.00** | 92.76 | **98.94** | **99.08** | 0.64 | 92.66 | **97.57** | **97.40** | 7.90 | 95.80 |
| OE | Yes | 99.60 | 99.61 | 0.51 | 92.61 | 98.39 | 96.24 | 4.34 | 92.30 | 95.39 | 92.53 | 17.72 | 95.72 |
| Energy FT | Yes | 98.83 | 99.14 | 1.31 | **92.79** | 98.68 | 96.82 | 2.84 | 92.52 | 96.61 | 94.92 | 13.78 | 94.83 |
| GNNSafe++ | Yes | 99.82 | 99.89 | 0.00 | 92.22 | 99.64 | 99.68 | **0.13** | 92.39 | **97.51** | **97.07** | **6.18** | **95.84** |
| +GeoEnergy | Yes | **99.96** | **99.98** | **0.00** | 92.38 | **99.80** | **99.75** | 0.18 | **92.68** | 97.19 | 96.71 | 8.66 | 95.76 |

Table 7: Performance of out-of-distribution detection using metrics such as AUROC(↑), AUPR(↑), and FPR95(↓), alongside in-distribution accuracy (ID ACC) on the Coauthur dataset across three types of OOD scenarios—structure manipulation, feature interpolation, and label leave-out—is presented. The best results are highlighted in bold.

| Model | OOD Expo | Coauthor-Structure | | | | Coauthor-Feature | | | | Coauthor-Label | | | |
|---|---|---|---|---|---|---|---|---|---|---|---|---|---|
| | | AUROC | AUPR | FPR95 | ID ACC | AUROC | AUPR | FPR95 | ID ACC | AUROC | AUPR | FPR95 | ID ACC |
| MSP | No | 95.30 | 94.37 | 24.75 | 92.47 | 97.05 | 96.93 | 15.55 | 92.45 | 94.88 | 97.99 | 23.81 | 95.18 |
| ODIN | No | 52.14 | 48.83 | 99.92 | 92.34 | 51.54 | 45.50 | 100.0 | 92.39 | 51.44 | 74.79 | 100.0 | 95.15 |
| Mahalanobis | No | 80.46 | 76.65 | 70.75 | 92.33 | 93.23 | 90.88 | 28.10 | 92.34 | 85.36 | 93.61 | 45.41 | 95.19 |
| Energy | No | 96.18 | 95.25 | 18.02 | 92.75 | 97.88 | 97.69 | 9.75 | 92.75 | 95.87 | 98.34 | 18.69 | 95.20 |
| GKDE | No | 65.87 | 72.65 | 99.48 | 88.62 | 80.69 | 86.47 | 96.57 | 84.72 | 61.15 | 81.39 | 94.60 | 89.05 |
| GPN | No | 34.67 | 40.21 | 99.57 | 89.45 | 81.77 | 80.56 | 74.46 | 87.05 | 93.24 | 97.55 | 34.78 | 91.68 |
| GNNSafe | No | 99.60 | 99.69 | 0.26 | **92.73** | 99.64 | 99.66 | 0.51 | **92.73** | 97.23 | 98.98 | 12.06 | 95.21 |
| +GeoEnergy | No | **99.92** | **99.86** | **0.15** | 92.10 | **99.83** | **99.78** | **0.51** | 92.60 | **97.90** | **99.23** | **9.48** | **95.37** |
| OE | Yes | 97.86 | 96.81 | 9.23 | 92.60 | 99.04 | 98.80 | 4.44 | 92.64 | 96.04 | 98.50 | 18.17 | 95.10 |
| Energy FT | Yes | 98.84 | 97.78 | 3.97 | 92.61 | 99.43 | 99.25 | 2.25 | 92.50 | 96.23 | 98.51 | 17.07 | 95.20 |
| GNNSafe++ | Yes | 99.99 | 99.99 | **0.02** | **92.92** | 99.97 | 99.95 | 0.09 | **92.87** | 97.89 | 99.24 | 9.43 | 95.24 |
| +GeoEnergy | Yes | **99.99** | **99.99** | 0.03 | 92.31 | **99.97** | **99.95** | **0.08** | 92.54 | **98.18** | **99.34** | **8.25** | **95.30** |

**GNNSafe and GNNSafe++ [25]:** GNNSafe combines energy-based models with graph neural networks to enhance OOD detection in graph data. It uses energy scores derived from node embeddings to differentiate between ID and OOD instances, leveraging graph structure for improved performance and robustness. GNNSafe++ extends GNNSafe by incorporating OOD data during training, fine-tuning the energy-based detection mechanism. This enhanced version uses OOD exposure to further improve the model's ability to distinguish between ID and OOD instances, achieving superior detection performance.

Table 8: Out-of-distribution detection performance measured by AUROC(↑)/AUPR(↑)/FPR95(↓) on OOD sub-graphs ES, FR and RU of **Twitch** dataset.

| Model | OOD Expo | Twitch-ES | | | Twitch-FR | | | Twitch-RU | | |
|---|---|---|---|---|---|---|---|---|---|---|
| | | AUROC | AUPR | FPR95 | AUROC | AUPR | FPR95 | AUROC | AUPR | FPR95 |
| MSP | No | 37.72 | 53.08 | 98.09 | 21.82 | 38.27 | 99.25 | 41.23 | 56.06 | 95.01 |
| ODIN | No | 83.83 | 80.43 | 33.28 | 59.82 | 64.63 | 92.57 | 58.67 | 72.58 | 93.98 |
| Mahalanobis | No | 45.66 | 58.82 | 95.48 | 40.40 | 46.69 | 95.54 | 55.68 | 66.42 | 90.13 |
| Energy | No | 38.80 | 54.26 | 95.70 | 57.21 | 61.48 | 91.57 | 57.72 | 66.68 | 87.57 |
| GKDE | No | 48.70 | 61.05 | 95.37 | 49.19 | 52.94 | 95.04 | 46.48 | 62.11 | 95.62 |
| GPN | No | 53.00 | 64.24 | 95.05 | 51.25 | 55.37 | 93.92 | 50.89 | 65.14 | 99.93 |
| GNNSafe | No | 49.07 | 57.62 | 93.98 | 63.49 | 66.25 | 90.80 | 87.90 | 89.05 | 43.95 |
| +GeoEnergy | No | **49.15** | **57.34** | **92.67** | **63.63** | **66.34** | **90.50** | **87.94** | **89.10** | **43.83** |
| OE | Yes | 55.97 | 69.49 | 94.94 | 45.66 | 54.03 | 95.48 | 55.72 | 70.18 | 95.07 |
| Energy FT | Yes | 80.73 | 87.56 | 76.76 | 79.66 | 81.20 | 76.39 | 93.12 | 95.36 | 30.72 |
| GNNSafe++ | Yes | 94.54 | 97.17 | 44.06 | 93.45 | 95.44 | 51.06 | **98.10** | **98.74** | 5.59 |
| +GeoEnergy | Yes | **94.90** | **97.34** | **41.33** | **93.54** | **95.49** | **50.65** | 98.07 | 98.69 | **5.56** |

# B Additional Experiment Results

## B.1 Additional OOD detection experiment results

**Further experiment results on the Amazon and Coauthor datasets.** In the main text, we succinctly present AUROC data for GeoEnergy across the Cora, Amazon, and Coauthor datasets. This appendix provides an in-depth exploration of those results, detailing comprehensive metrics including AUPR, FPR95, and ID ACC. As depicted in Table 6 and Table 7, extensive experimental outcomes for three distinct OOD perturbations—structure manipulation, feature interpolation, and label leave-out—are elaborated upon. These detailed analyses supplement the overview given in the main text, offering thorough insights into the performance across various models and experimental conditions. Notably, GeoEnergy consistently exhibits superior AUROC and AUPR scores under all scenarios, showcasing its robustness and efficacy in handling complex OOD challenges.

**Further experiment results on the Arxiv and Twitch datasets.** We provide average OOD detection performance results across sub-graphs for Arxiv and Twitch in main text. This appendix presents detailed results for specific sub-graphs within the Twitch and Arxiv datasets, demonstrating the effectiveness of GeoEnergy and its variants under varied scenarios. For the Twitch dataset, as shown in Table 8, detailed OOD detection performances are documented across three distinct sub-graphs (ES, FR, and RU). The table lists outcomes for different models under three types of OOD perturbations: structure manipulation, feature interpolation, and label leave-out. GeoEnergy consistently shows superior performance across all sub-graphs, particularly in reducing the FPR95, highlighting its capability to accurately identify OOD samples. Regarding the Arxiv dataset, Table 9 illustrates the performance for papers published in the years 2018, 2019, and 2020, treated as OOD samples. These results further validate the effectiveness of GeoEnergy in handling recent scientific publications.

**Summary of optimal hyperparameters for OOD detection.** As shown in Table 10, we provide a summary of the optimal hyperparameter settings for the scaling factor $s$ used in the GeoEnergy and +GeoEnergy across various datasets and OOD scenarios.

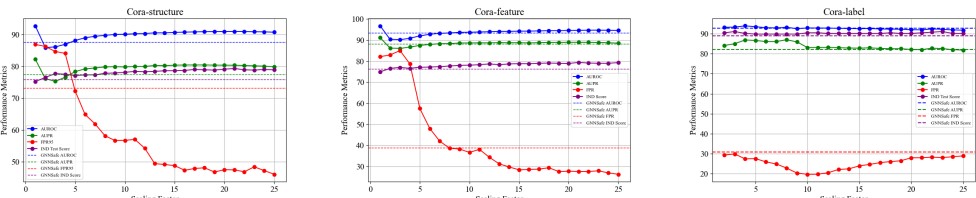

Figure 6: Impact of scaling factor variation on performance metrics for different OOD scenarios in graph neural networks on the Cora dataset.

**Sensitivity analysis of the scaling factor $s$ in GeoEnergy.** The three graphs in Figure 6 depict a sensitivity analysis of the scaling factor $s$ in GeoEnergy across different OOD scenarios (structure manipulation, feature interpolation, and label leave-out) on the Cora dataset. Each graph illustrates how performance metrics (AUROC, AUPR, FPR, and IND Test Score) evolve as the scaling factor varies from 1 to 25. AUROC and AUPR: Both metrics generally increase or stabilize with higher values of $s$, demonstrating improved classification and prediction precision under the influence of GeoEnergy. This suggests that increasing $s$ enhances the model's ability to distinguish between in-distribution and OOD samples, leading to more reliable predictions. FPR95: The false positive rate decreases significantly as $s$ increases, indicating that GeoEnergy effectively reduces the likelihood of false alarms in OOD detection. This is particularly evident in the sharp declines observed in the initial values of $s$. IND Test Score: The in-distribution accuracy remains relatively stable across different values of $s$, which implies that the model maintains its performance on seen data while improving its robustness to OOD samples.

## B.2 Expanded calibration results across different label rates

**Further calibration results across different label rates.** This section extends the examination of confidence calibration performance by providing detailed results for a broader range of label rates, specifically $L/C = 20, 40, 60$. Table 11 provides a comprehensive comparison of ECE

Table 9: Out-of-distribution detection performance measured by AUROC(↑)/AUPR(↑)/FPR95(↓) on OOD datasets of papers published in 2018, 2019 and 2020, respectively, on **Arxiv**.

| Model | OOD Expo | Arxiv-2018 | | | Arxiv-2019 | | | Arxiv-2020 | | |
|---|---|---|---|---|---|---|---|---|---|---|
| | | AUROC | AUPR | FPR95 | AUROC | AUPR | FPR95 | AUROC | AUPR | FPR95 |
| MSP | No | 61.66 | 70.63 | 91.67 | 63.07 | 66.00 | 90.82 | 67.00 | 90.92 | 89.28 |
| ODIN | No | 53.49 | 63.06 | 100.0 | 53.95 | 56.07 | 100.0 | 55.78 | 87.41 | 100.0 |
| Mahalanobis | No | 57.08 | 65.09 | 93.69 | 56.76 | 57.85 | 94.01 | 56.92 | 85.95 | 95.01 |
| Energy | No | 61.75 | 70.41 | 91.74 | 63.16 | 65.78 | 90.96 | 67.70 | 91.15 | 89.69 |
| GKDE | No | 56.29 | 66.78 | 94.31 | 57.87 | 62.34 | 93.97 | 60.79 | 88.74 | 93.31 |
| GPN | No | - | - | - | - | - | - | - | - | - |
| GNNSafe | No | 66.47 | 74.99 | 89.44 | 68.36 | 71.57 | 88.02 | 78.35 | 94.76 | 83.57 |
| +GeoEnergy | No | **67.30** | **75.70** | **87.88** | **69.33** | **72.45** | **86.20** | **79.33** | **95.01** | **80.48** |
| OE | Yes | 67.72 | 75.74 | 86.67 | 69.33 | 72.15 | 85.52 | 72.35 | 92.57 | 83.28 |
| Energy FT | Yes | 69.58 | 76.31 | 82.10 | 70.58 | 72.03 | 81.30 | 74.53 | 93.08 | 78.36 |
| GNNSafe++ | Yes | 70.40 | 78.62 | 81.47 | 72.16 | 75.43 | 79.33 | 81.75 | 95.57 | 71.50 |
| +GeoEnergy | Yes | **71.23** | **79.10** | **78.51** | **73.12** | **76.16** | **75.93** | **82.83** | **95.82** | **66.94** |

Table 10: Optimal hyperparameter selection for the scaling factor $s$ in GNNSafe(+GeoEnergy) and GNNSafe++(+GeoEnergy) across multiple datasets and OOD types.

| Model | Cora | | | Amazon | | | Coauthor | | | Twitch | Arxiv |
|---|---|---|---|---|---|---|---|---|---|---|---|
| | Structure | Feature | Label | Structure | Feature | Label | Structure | Feature | Label | | |
| GNNSafe(+GeoEnergy) | 20 | 15 | 15 | 20 | 20 | 10 | 20 | 5 | 5 | 1.5 | 1.5 |
| GNNSafe++(+GeoEnergy) | 20 | 15 | 2 | 20 | 5 | 2 | 5 | 5 | 1.5 | 1.2 | 1.5 |

for GCN, GAT, and GraphSage models across various datasets. The ECE values are recorded under several calibration techniques: uncalibrated (Uncal), temperature scaling (TS), matrix scaling (MS), and several methods specifically developed for GNN architectures including CaGCN, GCL, and our method, GeoEnergy. Each method's performance is scrutinized at different label rates, showcasing the robustness and consistency of GeoEnergy's superior calibration capability. The results highlight GeoEnergy's consistent outperformance across different settings and models, affirming its effectiveness in refining the calibration process, thereby enhancing the models' predictive accuracy and reliability across diverse scenarios. The detailed numerical results underscore the methodological advancements facilitated by GeoEnergy, reinforcing its role in improving both in-distribution and out-of-distribution detection by providing more accurate and reliable pseudo labels for effective self-training.

**Summary of optimal hyperparameters for confidence calibration across various scenarios.** We provide detailed hyperparameter settings for model calibration across various datasets and models, specifically GCN, GAT, and GraphSAGE. These settings pertain to optimal values used for experiments, which were not specified in the main text due to space constraints. The table 12 presents the selected parameters for each dataset (Cora, Citeseer, Pubmed) at different label rates ($L/C$ of 20, 40, and 60), facilitating a clearer understanding of how these configurations influence model performance.

Table 11: ECE (M=20) of different calibration methods on GCN, GAT, and GraphSAGE for different datasets with label rate $L/C = 20, 40, 60$.

| Dataset | L/C | GCN | | | | | | GAT | | | | | | GraphSAGE | | | | | |
|---|---|---|---|---|---|---|---|---|---|---|---|---|---|---|---|---|---|---|---|
| | | Uncal. | TS | MS | CaGCN | GCL | Ours | Uncal. | TS | MS | CaGCN | GCL | Ours | Uncal. | TS | MS | CaGCN | GCL | Ours |
| Cora | 20 | 0.1347 | 0.0488 | 0.0414 | 0.0401 | 0.0394 | **0.0391** | 0.1558 | 0.0717 | 0.0544 | 0.0450 | 0.0444 | **0.0430** | 0.1037 | 0.0463 | 0.0371 | 0.0398 | - | **0.0360** |
| | 40 | 0.1134 | 0.0417 | 0.0372 | 0.0407 | 0.0371 | **0.0338** | 0.1340 | 0.0485 | 0.0491 | 0.0365 | 0.0356 | **0.0330** | 0.0847 | 0.0330 | 0.0418 | 0.0368 | - | **0.0281** |
| | 60 | 0.0937 | 0.0886 | 0.0366 | 0.0376 | 0.0353 | **0.0302** | 0.1201 | 0.1192 | 0.0396 | 0.0308 | 0.0258 | **0.0241** | 0.0803 | 0.0323 | 0.0332 | 0.0339 | - | **0.0312** |
| Citeseer | 20 | 0.1248 | 0.0641 | 0.0644 | 0.0595 | 0.0579 | **0.0539** | 0.1534 | 0.0916 | 0.0633 | 0.0572 | 0.0660 | **0.0552** | 0.1135 | 0.0808 | 0.0866 | 0.0691 | - | **0.0579** |
| | 40 | 0.0957 | 0.0601 | 0.0538 | 0.0545 | 0.0575 | **0.0504** | 0.1252 | 0.0797 | 0.0590 | 0.0532 | 0.0603 | **0.0511** | 0.0942 | 0.0657 | 0.0586 | 0.0544 | - | **0.0468** |
| | 60 | 0.0806 | 0.0559 | 0.0521 | 0.0546 | 0.0501 | **0.0480** | 0.1090 | 0.0648 | 0.0519 | 0.0527 | 0.0522 | **0.0513** | 0.0684 | 0.0506 | 0.0416 | 0.0497 | - | **0.0400** |
| Pubmed | 20 | 0.0586 | 0.0541 | 0.0476 | 0.0405 | 0.0394 | **0.0332** | 0.0835 | 0.0656 | 0.0501 | 0.0356 | 0.0417 | **0.0316** | 0.0338 | 0.0337 | 0.0342 | 0.0364 | - | **0.0259** |
| | 40 | 0.0444 | 0.0446 | 0.0436 | 0.0402 | 0.0395 | **0.0349** | 0.0869 | 0.0658 | 0.0539 | 0.0308 | 0.0309 | **0.0265** | 0.0310 | 0.0275 | 0.0327 | 0.0275 | - | **0.0271** |
| | 60 | 0.0445 | 0.0367 | 0.0318 | 0.0311 | 0.0310 | **0.0302** | 0.0993 | 0.0669 | 0.0483 | 0.0308 | 0.0304 | **0.0293** | 0.0280 | 0.0321 | 0.0315 | 0.0239 | - | **0.0219** |

Table 12: Summary of used parameters on model calibration in GCN, GAT and GraphSAGE.

| Dataset | Cora | | | Citseer | | | Pubmed | | |
|---|---|---|---|---|---|---|---|---|---|
| $L/C$ | GCN | GAT | GraphSage | GCN | GAT | GraphSage | GCN | GAT | GraphSage |
| 20 | 15 | 15 | 15 | 10 | 10 | 15 | 4 | 3 | 16 |
| 40 | 15 | 10 | 15 | 8 | 7 | 15 | 3 | 3 | 20 |
| 60 | 15 | 12 | 15 | 7 | 6 | 15 | 3 | 3 | 15 |

# C Related Work

## C.1 Out-of-distribution Detection

OOD detection is crucial for identifying and managing samples that diverge from the training data distribution. This field has been rigorously researched and implemented across various neural network architectures, including traditional DNNs and the more complex GNNs. Due to the inherent differences in their architectures and data processing mechanisms, the methodologies appropriate for OOD detection differ between these types of networks.

**OOD detection for NNs.** Two principal approaches have been developed to tackle OOD detection in neural networks: scoring-based methods and training-time regularization techniques. The first focuses on designing scoring functions that utilize various metrics to assess the outlierness of samples. Notable examples include maximum softmax probability (MSP) [19], OpenMax score [33], Mahalanobis distance-based score [21], ODIN score [20], and energy-based scores [23, 34]. These functions provide quantifiable measures of how likely a sample is to be anomalous based on model predictions. Training-Time regularization techniques modify the training process to enhance models' OOD detection abilities. Strategies such as encouraging uniform prediction distributions for outliers [35, 22] and promoting higher energy outputs for potential OOD samples [23, 36, 37] have been explored. These techniques, aligning with log-likelihood shaping theories, are inherently suitable for OOD detection. Efforts to further enhance OOD detection also include methods leveraging unlabeled data for outlier identification [38]. However, they typically target *i.i.d.* instances and often overlook the interdependence between data points.

**OOD detection for GNNs.** To tackle the complexities of graph data with inter-dependent nodes, OOD detection for GNNs is a burgeoning field. GraphDE [11] utilizes a variational approach combined with mixed generative models to identify distribution shifts and effectively down-weight outliers, enhancing OOD detection capabilities for new datasets. In node classification, techniques like Graph-based Kernel Dirichlet Distribution Estimation (GKDE) [24] and Graph Posterior Network (GPN) [12] employ Bayesian GNN models that effectively consider the inter-dependence among nodes. Additionally, GNNSafe [13] introduces an energy-based OOD discriminator that operates independently of specific GNN architectures, offering a versatile solution to OOD challenges in graph neural networks. Uniquely, our method GeoEnergy harnesses the crucial insights provided by the intrinsic properties of node embeddings, specifically the natural clustering within the feature space based on angular relationships. This method extends the softmax loss to angular similarity loss and constrains weight vectors to a hypersphere, optimizing the angles between features. By leveraging the inherent low-dimensional manifold structure of graph data, GeoEnergy enhances the discriminative power of GNNs for effective OOD detection.

## C.2 Confidence Calibration of GNNs

The objective of confidence calibration is to align predicted probabilities with the actual likelihood of correctness. Calibration methods fall into two categories: post-hoc adjustments and regularization techniques. Post-hoc methods, such as Temperature Scaling (TS)[29], recalibrate predictions using a temperature parameter learned from validation data. Regularization techniques, like Focal Loss[39], mitigate overconfidence by minimizing KL divergence, while approaches such as Mixup[40] and Label Smoothing[41] adjust label entropy for better calibration. For GNNs, traditional calibration methods often underperform due to their unique structure. CaGCN[9] adapt TS, and GCL[10] introduce end-to-end calibration techniques tailored for GNNs, leveraging graph topology and entropy regularization to improve calibration. A GNN $f_\theta$ is perfectly calibrated when the predicted confidence $\hat{p}_i$ equals the actual probability $p_i$ of correctly predicting node $i$, ideally satisfying:

$\mathbb{P}(\hat{y}_i = y_i \mid \hat{p}_i = p) = p, \forall p \in [0, 1]$. The Expected Calibration Error (ECE) quantifies deviations from this ideal.

**Underconfidence Leading to Blurred OOD Detection.** Recent studies have shown that commonly used GNNs typically exhibit an underconfidence tendency [9], where the predicted confidence scores are systematically lower than the true probabilities. While underconfidence can reduce the risk of overconfident incorrect predictions, it has negative implications for OOD detection. Specifically, underconfidence makes it difficult for GNNs to effectively distinguish between in-distribution and out-of-distribution data, as the model's confidence estimates are insufficient to provide clear discriminative evidence.

