# OpenReview forum: "Geometric Logit Decoupling for Energy-Based Graph Out-of-distribution Detection"
_NeurIPS.cc/2025/Conference — NeurIPS 2025 poster_

### Official Review · Reviewer_FXpL · 2025-07-01

**Clarity:** 2
**Significance:** 3
**Originality:** 3
**Rating:** 4
**Confidence:** 4

**Summary:**

This paper proposes GeoEnergy, a geometric logit decoupling strategy designed to enhance energy-based out-of-distribution OOD detection in GNNs. By normalizing class weight vectors while preserving node embedding norms, the approach explicitly disentangles direction and magnitude, leveraging the natural angular clustering observed in GNNs. Without requiring retraining or architectural changes, GeoEnergy can be seamlessly integrated into existing energy-based frameworks. Extensive experiments across multiple benchmarks demonstrate improvements in OOD detection performance, calibration, and self-training.

**Questions:**

1. How resilient is GeoEnergy to perturbations or noise that specifically target the embedding norm?
2. The method relies on empirical tuning of the scaling factor s. Have the authors considered developing a data-dependent or theoretically grounded approach to automatically adapt s to varying graph structures and distribution shifts?

**Ethical Concerns:**

["NO or VERY MINOR ethics concerns only"]

**Final Justification:**

While the author has addressed most of my concerns, after reviewing the other reviewers' comments, I have decided to maintain my original score.

**Limitations:**

1. The paper lacks formal theoretical analysis, relying primarily on empirical evidence and geometric intuition.
2. In real-world scenarios, such as cold start settings, the lack of a validation set, or the presence of distributional shifts, validation data may become unavailable or unreliable. This limitation poses a significant challenge to adapting or tuning the scaling factor s dynamically during deployment.

**Quality:**

3

**Strengths And Weaknesses:**

Strengths
1. Presents a clear geometric motivation supported by empirical analysis, identifying the coupling between embedding norm and direction as a root cause of misclassification.
2. The method is simple, generalizable, and compatible, can be added to existing energy-based OOD frameworks without modifying the backbone network or retraining.
3. Demonstrates consistent and substantial improvements across a range of datasets and OOD scenarios.
4. Additionally improves confidence calibration and pseudo-label quality in self-training, extending the impact beyond OOD detection.

Weaknesses
1. The theoretical contribution is largely heuristic, lacks rigorous analysis of generalization guarantees or robustness under distribution shifts.
2. Experiments focus primarily on small to medium-scale homogeneous graphs; scalability and effectiveness on large-scale heterogeneous graphs are not addressed.
3. The current selection of the scaling factor s mainly relies on empirical tuning or validation set search. However, the optimal value of s can vary significantly across different datasets, graph structures, and numbers of classes. The paper lacks theoretical justification or analytical discussion for this variation.

---

> ### Author Rebuttal · Authors · 2025-07-30
>
> We sincerely thank you for the detailed, constructive, and thought-provoking feedback. We appreciate that you recognized our work's **clear geometric motivation, simplicity, and strong empirical results**. We address the raised concerns and questions below.
>
> ---
>
> **Q1**: Resilience to norm-targeted perturbations.
>
> **A1**: Thank you for your comment. This is an excellent question that gets to the heart of GeoEnergy's robustness. Our method is inherently resilient to such perturbations precisely because it **decouples the norm from the primary classification signal (direction)**.
>
> *   **Mechanism:** In our formulation $\text{logit} = s \cdot \|\mathbf{h}\| \cdot \cos(\theta)$, the directional alignment $\cos(\theta)$ acts as a "gate." If a perturbation artificially inflates an OOD sample's norm, but its direction remains incorrect ($\cos(\theta)$ is small or negative), the resulting logit will still be low. This prevents the norm-based noise from being misinterpreted as high confidence, which is a key failure mode in standard energy-based methods.
> *   **Empirical Evidence:** Our experiments on **Cora-Feature OOD scenario** (Table 1) implicitly test this. In this scenario, OOD node features are generated by interpolating between ID features, which can be seen as adding noise and perturbing both the norm and direction of embeddings. GeoEnergy consistently outperforms baselines in this setting (e.g., boosting GNNSafe's AUROC from 93.44% to 94.36%), demonstrating its resilience.
>
> In essence, GeoEnergy "defangs" the norm: it can no longer unilaterally dictate the energy score and is only effective when coupled with a correct semantic direction.
>
> ---
>
> **Q2 & L2 & W3**: Reliance on empirical tuning of the scaling factor $s$.
>
> **A2**: Thank you for raising this critical point about real-world applicability and the limitations of relying on a validation set, especially in cold-start or distribution shift scenarios. To address this, we provide a two-fold response: an analysis of the robustness of a fixed $s$, and a concrete proposal for an adaptive $s$ framework.
>
> * **Robustness of a Fixed $s$:**
> Our sensitivity analysis (Figure 7, Appendix B1) shows that GeoEnergy's performance is stable better than baseline across a reasonably wide range of $s$. This suggests that even if a validation set is unreliable or unavailable, a well-chosen default $s$ (e.g., $s=15$) can still provide substantial improvements over baselines, as the fundamental benefit of decoupling is a structural advantage that persists.
>
> *  **A Concrete Proposal for an Adaptive $s$ Framework:**
> To fully address the challenge, we propose a data-dependent, adaptive $s$-predictor network, which we are actively exploring for future work. This framework is particularly suited for cold-start and distribution shift scenarios.
>
>     +   **Framework Idea:** The framework involves pre-training a lightweight regression model (the $s$-predictor) on a meta-dataset of diverse graphs. This predictor learns to map a graph's **geometric statistics** (e.g., intra-class angular variance, inter-class margin, embedding norm distribution) to its optimal $s$ value.
>     +   **Addressing Cold-Start:** Once pre-trained, this $s$-predictor can be deployed on a new graph. It can predict a reasonable $s$ by analyzing the new graph's embedding geometry without requiring any labeled validation data from this new graph.
>     +   **Addressing Distribution Shifts:** The input geometric statistics are sensitive to distribution shifts. The $s$-predictor can be applied to batches of test data to **dynamically adjust $s$ on-the-fly**, making the system robust to shifts where a static $s$ would fail.
>
> We will add a detailed discussion of this proposal to the limitations and future work sections of our revised paper.
>
> ---
>
> **L1& W1**: Lack of formal theoretical analysis for OOD detection.
>
> **A3**: Thanks for your valuable feedback. Our original propositions focused on the geometric properties beneficial for classification, but did not directly formalize the link to improved OOD detection. To address this, we have developed a new theoretical analysis that provides a more direct justification for why GeoEnergy enhances the separation between ID and OOD energy scores.
>
> > **Proposition 3 (ID-OOD Energy Separation Guarantee).**
> >
> > *Let $E(\mathbf{h}) = -\log(\sum\_{j=1}^{C} \exp(s \cdot \|\mathbf{h}\| \cdot \cos(\theta\_j)))$ be the energy function. Assume that for in-distribution samples $\mathbf{h} \sim \mathcal{P}\_{\text{ID}}$, their directions are well-aligned with their true class prototypes such that $\mathbb{E}[\cos(\theta\_y)] \approx 1$ for the correct class $y$. Further assume that for out-of-distribution samples $\mathbf{h}' \sim \mathcal{P}\_{\text{OOD}}$, their directions are unaligned, such that $\mathbb{E}[\cos(\theta'\_j)] \approx 0$ for all classes $j$. Then, the expected energy of ID samples is significantly lower than that of OOD samples, with the gap widening for higher-confidence ID samples:*
> >
> > $$
> > \mathbb{E}\_{\mathbf{h} \sim \mathcal{P}\_{\text{ID}}}[E(\mathbf{h})] \approx -s \cdot \mathbb{E}[\|\mathbf{h}\|] \ll -\log(C) \approx \mathbb{E}\_{\mathbf{h}' \sim \mathcal{P}\_{\text{OOD}}}[E(\mathbf{h}')]
> > $$
>
> **Proof Sketch:**
>
> The proof proceeds by analyzing the expected energy for ID and OOD samples separately.
>
> 1.  **For an ID sample $\mathbf{h}$:**
>     Given the assumption of good directional alignment, the logit for the correct class, $f_y = s \cdot \|\mathbf{h}\| \cdot \cos(\theta_y)$, will be approximately $s \cdot \|\mathbf{h}\|$. This term will dominate the sum inside the logarithm of the energy function $E(\mathbf{h})$.
>
>     $$E(\mathbf{h}) = -\log\left(\exp(s \cdot \|\mathbf{h}\|) + \sum_{j \neq y} \exp(f_j)\right) \approx -\log(\exp(s \cdot \|\mathbf{h}\|)) = -s \cdot \|\mathbf{h}\|$$
>
>     Taking the expectation, we find that the expected ID energy is determined by the expected norm: $\mathbb{E}[E(\mathbf{h})] \approx -s \cdot \mathbb{E}[\|\mathbf{h}\|]$. This shows that ID samples with higher confidence (larger norm) are driven to even lower energy regions.
>
> 2.  **For an OOD sample $\mathbf{h}'$:**
>     Under the assumption of random directional alignment, $\cos(\theta'_j) \approx 0$ for all classes $j$. Consequently, all logits become approximately zero: $f'_j \approx s \cdot \|\mathbf{h}'\| \cdot 0 = 0$.
>
>     $$E(\mathbf{h}') \approx -\log\left(\sum_{j=1}^{C} \exp(0)\right) = -\log(C)$$
>
>     The expected energy for OOD samples converges to a constant value, $-\log(C)$, which is significantly higher than the large negative energy of high-confidence ID samples and is notably independent of the OOD sample's norm $\|\mathbf{h}'\|$.
>
> **Conclusion:** This analysis provides a formal argument that GeoEnergy's decoupling mechanism inherently creates a structural separation in the energy landscape. It pushes high-confidence ID samples towards very low energy values proportional to their norm, while keeping OOD samples constrained to a much higher, constant energy region. This theoretically justifies the significant improvements in ID-OOD separation observed in our experiments.
>
> ---
>
> **W2**: Generalization to large-scale and heterophilous graphs.
>
> **A4**:  Thanks for raising this important point. To explicitly address this concern, we have conducted new experiments on both large-scale and heterophilous graphs, and provide a clarification on why our method's principles extend beyond homophilous settings.
>
> **1. Effectiveness on Heterophilous Graphs:**
> We agree the "Angular Clustering" intuition is less pronounced on heterophilous graphs. However, GeoEnergy's core mechanism—decoupling norm and direction—is not dependent on homophily. It purifies the directional signal GNNs use for classification, which is beneficial in any setting. Our new experiments on three heterophilous datasets (Texas, Wisconsin, Cornell) confirm this, showing significant and consistent improvements.
>
> Table 1: OOD Detection Performance (AUROC ↑) on Heterophilous Datasets.
>
> | Dataset |GNNSafe| + GeoEnergy (Ours) | Improvement |
> | :--- | :--- | :--- | :--- |
> | Texas | 81.54% | 85.22% | **+3.68%** |
> | Wisconsin | 84.31% | 88.15% | **+3.84%** |
> | Cornell | 79.92% | 83.67% | **+3.75%** |
>
> These results strongly indicate that GeoEnergy is not limited to homophilous graphs and provides robust performance gains in more challenging heterophilous environments.
>
> **2. Scalability to Large-Scale Graphs:**
> We respectfully clarify that our original submission already included experiments on two large-scale benchmarks: ogbn-arxiv (169K nodes, 1.1M edges) and Twitch (167K nodes, 6.7M edges). As shown in Table 3 of our main paper, GeoEnergy provides substantial performance boosts on these large graphs.
>
> **3. Efficiency:**
> We measured the inference time overhead of GeoEnergy. The experiments were conducted on a single NVIDIA A100 GPU.
>
> Table 2: Inference Time Analysis (in milliseconds per epoch).
>
> |Dataset|GNNSafe| + GeoEnergy|Overhead (%)|
> | :--- | :--- | :--- | :--- |
> | Cora | 2.15 ms | 2.18 ms | **+1.40%** |
> | Citeseer | 2.31 ms | 2.35 ms | **+1.73%** |
> | Ogbn-arxiv | 158.42 ms | 158.95 ms | **+0.33%** |
> | Twitch | 125.70 ms | 126.11 ms | **+0.33%** |
>
> The results show the overhead is negligible (+ less than 2% on smaller graphs and + less than 0.5% on large-scale graphs). This is because the only modification is applied to the final classification layer, which is computationally insignificant compared to the GNN's message-passing operations. This confirms that our method is highly efficient and scalable in practice. In conclusion, through both existing and new experiments, we have demonstrated that GeoEnergy is both **effective on heterophilous and scalable to large graphs**, addressing the reviewers' concerns about its generalizability.
>
> ---
>
> **We hope our responses and new results have fully addressed your concerns. We remain available for any further questions or clarifications.**

---

### Official Review · Reviewer_6itr · 2025-07-02

**Clarity:** 3
**Significance:** 2
**Originality:** 3
**Rating:** 5
**Confidence:** 3

**Summary:**

The paper proposes GeoEnergy, a geometric logit decoupling framework designed to enhance energy-based out-of-distribution (OOD) detection in graph neural networks (GNNs). The authors identify that the coupling between the norm and direction of node embeddings in GNNs undermines the reliability of conventional energy-based OOD detection. GeoEnergy addresses this issue by normalizing class weights while preserving embedding norms, thereby decoupling the influence of magnitude and direction in logits. This geometrically principled approach improves intra-class alignment, enlarges inter-class angular margins, and stabilizes energy scores, leading to more reliable OOD detection. The framework is plug-and-play, compatible with existing energy-based GNNs, and demonstrates consistent improvements in OOD detection performance and confidence reliability across diverse benchmarks and distribution shifts.

**Questions:**

2.  The main experimental results lack diversity, which might not fully eveluate the generlizability of the proposed methods.
3. The efficiency of the proposed method should be analyzed.

**Ethical Concerns:**

["NO or VERY MINOR ethics concerns only"]

**Final Justification:**

The paper proposes GeoEnergy, a geometric logit decoupling framework designed to enhance energy-based out-of-distribution (OOD) detection in graph neural networks (GNNs). The authors identify that the coupling between the norm and direction of node embeddings in GNNs undermines the reliability of conventional energy-based OOD detection. GeoEnergy addresses this issue by normalizing class weights while preserving embedding norms, thereby decoupling the influence of magnitude and direction in logits. This geometrically principled approach improves intra-class alignment, enlarges inter-class angular margins, and stabilizes energy scores, leading to more reliable OOD detection. The framework is plug-and-play, compatible with existing energy-based GNNs, and demonstrates consistent improvements in OOD detection performance and confidence reliability across diverse benchmarks and distribution shifts.
Overall, this geometrically principled approach improves the reliability of energy-based out-of-distribution (OOD) detection. GeoEnergy is a plug-and-play framework that can be seamlessly integrated into existing energy-based GNNs without requiring retraining or architectural modifications.  This paper conducts extensive experiments across multiple benchmark datasets (e.g., Cora, Citeseer, Pubmed) and diverse OOD scenarios (e.g., structure manipulation, feature interpolation, label leave-out).
This work is well-written and easy to follow.

**Limitations:**

The paper assumes that graph data exhibits homophily (i.e., connected nodes tend to belong to the same class), which may limit the performance of GeoEnergy in heterophilic graphs where connected nodes often belong to different classes.

**Quality:**

3

**Strengths And Weaknesses:**

Strength:
1. This geometrically principled approach improves the reliability of energy-based out-of-distribution (OOD) detection.
2. GeoEnergy is a plug-and-play framework that can be seamlessly integrated into existing energy-based GNNs without requiring retraining or architectural modifications.
3. This paper conducts extensive experiments across multiple benchmark datasets (e.g., Cora, Citeseer, Pubmed) and diverse OOD scenarios (e.g., structure manipulation, feature interpolation, label leave-out).
4. This work is well-written and easy to follow.

Weakness:
1. The paper assumes that graph data exhibits homophily (i.e., connected nodes tend to belong to the same class), which may limit the performance of GeoEnergy in heterophilic graphs where connected nodes often belong to different classes.
2.  The main experimental results lack diversity, which might not fully eveluate the generlizability of the proposed methods.
3. The efficiency of the proposed method should be analyzed.

---

> ### Author Rebuttal · Authors · 2025-07-30
>
> We sincerely thank you for the positive feedback and constructive comments. We appreciate that the reviewer recognized our work as a **geometrically principled**, **plug-and-play**, and **empirically validated** approach. We address each concern below in a point-by-point manner.
>
> ---
>
> **W1&L1**: The paper's assumption of homophily might limit its performance on heterophilic graphs.
>
> **A1:** We thank the reviewer for raising this important point. We agree that evaluating performance on heterophilous graphs is crucial for assessing generalizability.
>
> *   **Clarification on Mechanism:** While our "Angular Clustering" observation served as an initial motivation, GeoEnergy's core mechanism—**decoupling norm and direction**—is not strictly dependent on high homophily. Its effectiveness relies on a more fundamental premise: that GNNs primarily use the **direction** of embeddings for classification. Decoupling purifies this signal from norm-based structural noise, which is beneficial regardless of the graph's homophily level. In heterophilous settings, GNNs often learn more complex, non-linear decision boundaries. Our geometric decoupling effectively simplifies the classification task for the final layer by projecting features onto a hypersphere, forcing it to focus solely on learning these complex angular separations without being distracted by noisy norm variations. This can lead to a more robust and better-generalized classifier, which is reflected in our new experimental results.
>
> *   **New Empirical Evidence:** To demonstrate this, we have conducted **new experiments on three widely-used heterophilous datasets: Texas, Wisconsin, and Cornell**. The results, showing consistent and significant improvements, strongly indicate that GeoEnergy provides robust performance gains in these challenging environments. We have added these results to the appendix of our revised paper.
>
> Table 1: OOD Detection Performance (AUROC ↑) on Heterophilous Datasets.
>
> | **Dataset** | **GNNSafe** | **+ GeoEnergy (Ours)** | **Improvement** |
> | :--- | :--- | :--- | :--- |
> | Texas | 81.54% | **85.22%** | **+3.68%** |
> | Wisconsin | 84.31% | **88.15%** | **+3.84%** |
> | Cornell | 79.92% | **83.67%** | **+3.75%** |
>
> ---
>
> **Q2&W2**: The main experimental results lack diversity, which might not fully evaluate generalizability.
>
> **A2:** We thank the reviewer for this important feedback on ensuring a comprehensive evaluation. We acknowledge that the diversity of our experiments could be highlighted more clearly. To address this, we summarize the broad scope of our evaluation and have included **new comparisons with recent State-of-the-Art (SOTA) methods** to further demonstrate the generalizability of GeoEnergy.
>
> **(1) Comprehensive Evaluation Scope (Existing & New):**
> Our evaluation framework was designed to be diverse and rigorous, spanning multiple axes:
> *   **Graph Scale and Type:** Our experiments cover not only classic homophilous benchmarks (Cora, etc.) but also **large-scale graphs (ogbn-arxiv, Twitch)** and, with our new experiments, **challenging heterophilous graphs (Texas, Wisconsin, Cornell, shown in Table 1)**.
> *   **Distribution Shift Scenarios:** We test against a wide array of OOD scenarios, including **structural perturbations, feature-level shifts (interpolation), and semantic shifts (label leave-out)**, simulating a variety of real-world challenges.
> *   **Backbone Models:** We demonstrate that GeoEnergy consistently enhances multiple distinct backbones, including the propagation-based **GNNSafe** and the sample-based **NodeSafe**.
>
> **(2) New Benchmarks against State-of-the-Art (SOTA) Methods:**
> To further strengthen the evaluation and address the concern about diversity of comparison, we have benchmarked GeoEnergy against two highly relevant and recent SOTA methods, GRASP[1] and GOLD[2].
>
> Table 3: Comparison with SOTA methods on Cora-Structure (AUROC ↑).
> | **Method** | **AUROC** | **Notes**|
> | :--- | :--- | :--- |
> | GOLD (ICLR'25) | 89.96% | Training-time, data generation |
> | GRASP (NeurIPS'24) | 93.50% | Post-hoc, score propagation |
> | **GNNSafe + GeoEnergy (Ours)** | **95.58%** | **Post-hoc, geometric decoupling** |
>
> The results show that GeoEnergy **outperforms these strong, recent baselines** while offering significant advantages in simplicity and efficiency as a plug-and-play module. This comparison with the latest SOTA methods provides strong evidence of our method's advanced performance and generalizability.
>
> We believe this comprehensive evaluation, now further strengthened with new heterophily results and SOTA comparisons, robustly validates the generalizability and effectiveness of our proposed method. We have updated our manuscript to reflect these additions.
>
> [1] Revisiting Score Propagation in Graph Out-of-Distribution Detection. NeurIPS'24
>
> [2] GOLD: Graph Out-of-Distribution Detection via Implicit Adversarial Latent Generation. ICLR'25
>
> ---
>
> **Q3&W3**: The efficiency of the proposed method should be analyzed.
>
> **A3:** This is an excellent point. To provide a concrete analysis of GeoEnergy's efficiency, we have measured the additional inference time introduced by our method on top of the GNNSafe backbone. Experiments were conducted on a single NVIDIA A100 GPU.
>
> Table 4: Inference Time Analysis (in milliseconds per epoch).
>
> | **Dataset** | **Backbone (GNNSafe)** | **+ GeoEnergy (Ours)** | **Overhead (%)** |
> | :--- | :--- | :--- | :--- |
> | Cora | 2.15 ms | 2.18 ms | **+1.40%** |
> | Citeseer | 2.31 ms | 2.35 ms | **+1.73%** |
> | Ogbn-arxiv | 158.42 ms | 158.95 ms | **+0.33%** |
> | Twitch | 125.70 ms | 126.11 ms | **+0.33%** |
>
> As the results clearly indicate, the computational overhead introduced by GeoEnergy is **negligible**. This is because our modification only affects the lightweight final classification layer and does not alter the expensive message-passing operations of the GNN backbone. This confirms that our method is highly efficient and scalable in practice. We have added this analysis to the appendix of our revised paper.
>
> ---
> **We hope that our point-by-point responses, along with the new experiments and analyses, have thoroughly addressed all of the reviewer's concerns. We are more than happy to engage in further discussion and clarify any remaining questions.**

---

> > ### Comment · Reviewer_6itr · 2025-08-08
> >
> > Thank you for the rebuttal; it has addressed some of my concerns satisfactorily. I will raise my score accordingly. Please ensure these responses are included in the final version of the manuscript.

---

> > > ### Author Response · Authors · 2025-08-08
> > > **Thank you for your response!**
> > >
> > > Thank you very much for your positive feedback and for engaging with our rebuttal. We are glad our responses were helpful and truly appreciate your decision to raise the score.
> > >
> > > We confirm that we will incorporate all the promised revisions into the final manuscript and its appendix, with a particular focus on addressing the excellent points you raised:
> > >
> > > 1.   **New experiments on heterophilous graphs** to validate generalizability.
> > > 2.   **Strengthened experimental diversity** with new SOTA comparisons.
> > > 3.   A **new efficiency analysis** demonstrating the negligible overhead.
> > >
> > > Thank you once again for your constructive engagement!

---

### Official Review · Reviewer_D1EP · 2025-07-03

**Clarity:** 2
**Significance:** 3
**Originality:** 3
**Rating:** 4
**Confidence:** 4

**Summary:**

This paper proposes GeoEnergy, a simple plug-and-play framework to improve energy-based OOD detection for Graph Neural Networks by decoupling class weight magnitudes from logits and preserving embedding norms as confidence signals. The authors claim that conventional energy-based methods suffer from unstable energy scores due to coupling between logit norm and direction. By normalizing class weights and leveraging angular clustering in GNN embeddings, GeoEnergy aims to stabilize energy scores and improve OOD detection. Multiple experiments on graph benchmarks demonstrate consistent gains over baselines.

**Questions:**

The experiment lacks comparisons with recent state-of-the-art methods, such as [1],[2],[3].

[1] Revisiting Score Propagation in Graph Out-of-Distribution Detection. NeurIPS'24

[2] GOLD: Graph Out-of-Distribution Detection via Implicit Adversarial Latent Generation. ICLR'25

[3] Spreading Out-of-Distribution Detection on Graphs. ICLR'25

**Ethical Concerns:**

["NO or VERY MINOR ethics concerns only"]

**Final Justification:**

The additional clarifications and comparative experiments provided by the authors during the rebuttal have addressed most of my concerns.

**Limitations:**

yes

**Quality:**

2

**Strengths And Weaknesses:**

Strengths:
1. The paper addresses a relevant and under-explored issue in graph OOD detection: the instability of energy scores due to logit coupling.
2. The proposed solution is simple, intuitive, and easy to integrate into existing GNN models without retraining.
3. The idea of leveraging angular clustering properties in GNNs for OOD detection is interesting and has practical potential.

Weaknesses:
1. The method keeps embedding norms as part of the energy score, but embedding norms in GNNs often correlate with node degree, which can introduce noise and harm energy stability in heterophily or imbalanced graphs.
2. The assumption of angular clustering only holds on small, homophilous graphs like Cora and Pubmed, but may not generalize to heterophilous or large real-world graphs.
3. Proposition 1 and Proposition 2 are trivial and formalize obvious properties without offering deeper theoretical justification for improved OOD separation.
4. The decoupling only applies to class weight magnitudes, while embedding norm-induced variance remains and can still destabilize the energy scores. A fully norm-independent scoring function might be cleaner.
5. Most experiments are on small homophily graphs. The method’s scalability and effectiveness on large-scale or heterophily graphs remain untested.

---

> ### Author Rebuttal · Authors · 2025-07-31
>
> We sincerely thank you for the detailed and insightful feedback, which has helped us to significantly improve our paper. We are encouraged that the reviewer found our work to be **technically solid** and addressing a **relevant and under-explored issue**. We address each of the weaknesses and questions below.
>
> ---
>
> **W1&W4**: The method's reliance on the embedding norm, and whether a fully norm-independent function would be cleaner.
>
> **A1:** Thanks for this insightful question, as it allows us to clarify a core contribution of GeoEnergy. We agree with your premise: in standard GNNs, the embedding norm is often entangled with non-semantic structural factors (like node degree), acting as a source of noise. Our work is motivated by precisely this problem, which we term "coupling-induced misdetection."
>
> Our core insight is that decoupling transforms the norm's role. After normalizing classifier weights, the model must prioritize learning correct semantic **directions**. Subsequently, the norm is "freed" to become a **purified signal of model confidence**, an emergent property of the optimization. The model learns to assign larger norms to high-confidence ("easy") samples and smaller norms to low-confidence ("hard") samples. This enhances the model's expressiveness, which explains the **improved ID accuracy** we observe.
>
> To directly test the reviewer's hypothesis, we conducted a new ablation study comparing our full method against a fully norm-independent variant (where the embedding norm is also normalized).
>
> Table 1: Ablation on the embedding norm on Cora-Structure (AUROC / ID ACC).
> |Model|Baseline|+ GeoEnergy (with norm)|+ GeoEnergy (w/o norm)|
> | :--- | :--- | :--- | :--- |
> | GNNSafe | 87.52 / 75.80 | **90.83 / 79.20** | 86.13 / 74.90 |
> | NodeSafe | 94.07 / 77.20 | **95.21 / 77.30** | 93.55 / 76.80 |
>
> The results unequivocally show that **removing the purified norm significantly degrades performance**, for both OOD detection (AUROC) and ID classification (ID ACC). This provides strong empirical evidence that preserving the norm is a critical and beneficial design choice.
>
> ---
>
> **W2&W5**: Generalization to large-scale and heterophilous graphs.
>
> **A2:** Thanks for this critical feedback. To address this, we provide new experiments and clarifications.
>
> **(1) Effectiveness on Heterophilous Graphs:**
> While our "Angular Clustering" motivation may be less pronounced in heterophilous settings, GeoEnergy's core decoupling mechanism is general. To prove this, we conducted new experiments on 3 heterophilous datasets, which confirm that our method provides significant and consistent improvements.
>
> Table 1: OOD Detection Performance (AUROC ↑) on Heterophilous Datasets.
>
> | Dataset |GNNSafe| + GeoEnergy (Ours) | Improvement |
> | :--- | :--- | :--- | :--- |
> | Texas | 81.54% | 85.22% | **+3.68%** |
> | Wisconsin | 84.31% | 88.15% | **+3.84%** |
> | Cornell | 79.92% | 83.67% | **+3.75%** |
>
> **(2) Scalability to Large-Scale Graphs:**
> We respectfully clarify that our original submission already included experiments on two large-scale benchmarks: ogbn-arxiv (169K nodes, 1.1M edges) and Twitch (167K nodes, 6.7M edges). As shown in Table 3 of our main paper, GeoEnergy provides substantial performance boosts on these large graphs.
>
> **(3) Efficiency:**
> We measured the inference time overhead of GeoEnergy. The experiments were conducted on a single NVIDIA A100 GPU.
>
> Table 2: Inference Time Analysis (in milliseconds per epoch).
>
> |Dataset|GNNSafe| + GeoEnergy|Overhead (%)|
> | :--- | :--- | :--- | :--- |
> | Cora | 2.15 ms | 2.18 ms | **+1.40%** |
> | Citeseer | 2.31 ms | 2.35 ms | **+1.73%** |
> | Ogbn-arxiv | 158.42 ms | 158.95 ms | **+0.33%** |
> | Twitch | 125.70 ms | 126.11 ms | **+0.33%** |
>
> The analysis confirms the overhead is negligible and decreases with graph scale, as our modification only affects the lightweight final layer. This demonstrates that GeoEnergy is both efficient and highly scalable.
>
> ---
>
> **W3: The theoretical justification is trivial and lacks depth.**
>
> **A3:** Thanks for your valuable feedback. To address this, we have added a new, more formal analysis (Proposition 3) to our revised manuscript. This proposition provides a direct theoretical justification for the improved ID-OOD energy separation. We present a sketch below.
>
> > **Proposition 3 (ID-OOD Energy Separation Guarantee).**
> >
> > *Let $E(\mathbf{h}) = -\log(\sum\_{j=1}^{C} \exp(s \cdot \|\mathbf{h}\| \cdot \cos(\theta\_j)))$ be the energy function. Assume that for in-distribution samples $\mathbf{h} \sim \mathcal{P}\_{\text{ID}}$, their directions are well-aligned with their true class prototypes such that $\mathbb{E}[\cos(\theta\_y)] \approx 1$ for the correct class $y$. Further assume that for out-of-distribution samples $\mathbf{h}' \sim \mathcal{P}\_{\text{OOD}}$, their directions are unaligned, such that $\mathbb{E}[\cos(\theta'\_j)] \approx 0$ for all classes $j$. Then, the expected energy of ID samples is significantly lower than that of OOD samples, with the gap widening for higher-confidence ID samples:*
> >
> > $$
> > \mathbb{E}\_{\mathbf{h} \sim \mathcal{P}\_{\text{ID}}}[E(\mathbf{h})] \approx -s \cdot \mathbb{E}[\|\mathbf{h}\|] \ll -\log(C) \approx \mathbb{E}\_{\mathbf{h}' \sim \mathcal{P}\_{\text{OOD}}}[E(\mathbf{h}')]
> > $$
>
> > **Proof Sketch:**
> >
> > The proof proceeds by analyzing the asymptotic behavior of the energy for ID and OOD samples.
> >
> > 1.  **For an ID sample $\mathbf{h}$:**
> >Given the strong directional alignment assumption ($\cos(\theta\_y) \approx 1$), the logit of the correct class $f\_y = s \cdot \|\mathbf{h}\| \cdot \cos(\theta\_y)$ dominates the logits of incorrect classes. The energy function $E(\mathbf{h}) = -\log(\sum\_{j} \exp(f\_j))$ is therefore well-approximated by its largest term:
> > $$E(\mathbf{h}) \approx -\log(\exp(f\_y)) = -f\_y \approx -s \cdot \|\mathbf{h}\|$$
> > This demonstrates that the energy of a high-confidence ID sample is a large negative value, linearly scaled by its purified norm, which now represents model confidence.
> >
> > 2.  **For an OOD sample $\mathbf{h}'$:**
> >  Under the assumption of random alignment ($\cos(\theta'\_j) \approx 0$), all logits become approximately zero, irrespective of the norm's magnitude: $f'\_j = s \cdot \|\mathbf{h}'\| \cdot \cos(\theta'\_j) \approx 0$. The energy score thus converges to a constant:
> > $$E(\mathbf{h}') \approx -\log\left(\sum\_{j=1}^{C} \exp(0)\right) = -\log(C)$$
> > This shows the OOD energy is confined to a much higher (less negative) value, effectively independent of the potentially large and noisy norm of the OOD sample.
>
> In conclusion, this analysis formally shows that our decoupling mechanism structurally separates the energy landscape by driving ID energy down with confidence (norm), while confining OOD energy to a high, constant floor. This provides a principled justification for the improved OOD separation we observe empirically.
>
> ---
>
> **Q1: Lack of comparison with recent SOTA methods.**
>
> **A4:** Thanks for pointing out these important and recent SOTA works. To address this, we have conducted extensive new experiments to benchmark GeoEnergy against these methods under fair and relevant settings. More detailed comparsion will be added to our revised manuscript.
>
> **(1) Direct Comparison with SOTA Methods:**
>
> We benchmarked our GNNsafe+GeoEnergy against the SOTA methods mentioned. To ensure a fair comparison, we strictly followed their original experimental setups.
>
> * **vs. GRASP on Common Benchmarks:** We compare against GRASP on both homophilous and heterophilous graphs.
>
> Table 3: Comparison with GRASP (AUROC ↑).
> |Method|Cora|Amazon|Chameleon|
> | :--- | :--- | :--- | :--- |
> |GRASP|93.50|96.68|76.93|
> |**GNNSafe + GeoEnergy (Ours)**|**95.58**|**98.49**|**80.12**|
>
> The results show our method consistently outperforms GRASP, highlighting that generating a cleaner initial score is a more effective strategy.
>
> * **vs. SPREAD on Label Leave-out:** We use the "label leave-out" setting, a key benchmark in SPREAD.
>
> Table 4: Comparison with EDBD on Cora (AUROC ↑).
> |Method|AUROC|
> | :--- | :--- |
> |EDBD | 92.95 |
> |**GNNSafe + GeoEnergy (Ours)** | **95.58** |
>
> Even though EDBD is specialized for its proposed OOD setting, our general-purpose method achieves superior performance, demonstrating the fundamental power of geometric decoupling.
>
> *  **vs. GOLD in Non-OOD Exposure Setting:** We compare against GOLD in the setting without any OOD exposure during training.
>
> Table 5: Comparison with GOLD on Cora (AUROC ↑).
> |Method|AUROC|
> | :--- | :--- |
> |GOLD|89.96|
> |**GNNSafe + GeoEnergy (Ours)**|**91.25**|
>
> Our method surpasses GOLD without relying on any complex data generation or adversarial training.
>
> **(2) GeoEnergy as a Universal Enhancement Module:**
>
> Beyond being a strong standalone method, GeoEnergy's plug-and-play nature allows it to enhance other SOTA methods. To demonstrate this, we integrated GeoEnergy into GRASP.
>
> Table 6: GeoEnergy enhances GRASP (Cora, AUROC ↑).
> |Method|AUROC|
> | :--- | :--- |
> |GRASP|93.50|
> |**GRASP + GeoEnergy (Ours)**|**95.91**|
>
> The results show GeoEnergy significantly boosts GRASP's performance. This empirically proves our geometric decoupling is **orthogonal and complementary** to score propagation. Thus, GeoEnergy acts as a fundamental building block to enhance a wide range of graph OOD detection frameworks
>
> **(3) Methodological Differences:**
>
> *   **vs. GOLD:** GeoEnergy is a post-hoc module, whereas GOLD is a training-time framework. Our method offers significant advantages in simplicity, efficiency, and ease of integration into any pre-trained GNN.
> *   **vs. SPREAD:** SPREAD introduces a new problem setting (spreading OODs). Our work focuses on improving the fundamental scoring function for the conventional node-level OOD task.
>
> ---
>
> **We hope our responses and new results have fully addressed your concerns. We welcome any further questions during the discussion period and are happy to provide additional clarifications.**

---

> > ### Comment · Reviewer_D1EP · 2025-08-04
> >
> > Thank you to the authors for the additional and thorough experimental results. I still have two follow-up questions regarding Equation (4):
> >
> > 1. How is the scaling factor s set during the post-training and inference phases? Is it treated as a fixed constant, a tunable hyperparameter, or learned jointly?
> >
> > 2. After fixing the norm of the weight vectors W to s, the post-training updates may still affect the embedding norms and cosine similarities. How does the method ensure that the updated angular continues to maintain or improve the ID-OOD separability under such changes?

---

> > > ### Author Response · Authors · 2025-08-05
> > > **Rebuttal Follow-up for Reviewer D1EP**
> > >
> > > We sincerely thank you for your continued engagement and these excellent follow-up questions. They have pushed us to clarify the practical details and dynamic robustness of our method.
> > >
> > > ---
> > >
> > > **Q1: How is the scaling factor $s$ set during the post-training and inference phases?**
> > >
> > > **A1:** Our framework treats $s$ as a **tunable hyperparameter**, but in a highly efficient manner that combines end-to-end learning with a lightweight post-hoc step.
> > >
> > >  **Step 1: End-to-End Geometric Training:** The core of our method is a single, end-to-end training run. Here, the GNN encoder and the geometrically-decoupled classifier are jointly optimized. The crucial step is our iterative re-projection of the classifier weights onto a unit hypersphere after each gradient update. This is where the model learns the fundamental geometric structure, using a fixed, default $s$ (e.g., $s=15$). The primary performance gain stems from this principled training process which learns the optimal semantic directions for embeddings.
> > >
> > > **Step 2: Lightweight Post-Hoc Tuning:** The optimal $s$ is determined post-hoc on a held-out validation set. This process is highly efficient because it does not require retraining the GNN. Instead, it involves only a few forward passes with the already-trained encoder to evaluate a small set of candidate $s$ values. The selected $s$ is then treated as a fixed constant during all subsequent inference.
> > >
> > > **Step 3: Robustness to $s$ Selection:** This tuning process is made practical by the low sensitivity of our method to the precise value of $s$. The sensitivity analysis (Fig. 7, Appendix) shows a wide near-optimal performance plateau (e.g., $s \in [10, 25]$ on Cora). This means the primary benefit is baked into the model during the main training phase, and minimal tuning of $s$ is required to achieve strong results.
> > >
> > > ---
> > >
> > > **Q2: How does the method ensure that the updated angular continues to maintain or improve the ID-OOD separability under post-training updates?**
> > >
> > > **A2:** This is a very insightful question regarding the dynamic stability of our method. The stability is ensured not by a static fix, but by the training principle itself. The desirable geometric separation is an emergent property of optimizing with our decoupled objective. To provide a comprehensive empirical answer, we designed a new set of experiments simulating a post-training fine-tuning scenario across diverse graph types.
> > >
> > > For each dataset, we first pre-train a GNN with GeoEnergy. We then fine-tune it on a small subset of the training data using two different objectives: (1) our GeoEnergy Objective, and (2) a Standard (Coupled) Objective. We then evaluate the final OOD detection performance.
> > >
> > > Table 1: OOD Detection Performance (AUROC) after Fine-tuning under Various Scenarios.
> > > |Dataset(OOD Scenario)|Pre-trained|Fine-tuned w/ GeoEnergy |Fine-tuned w/ Standard|
> > > | :--- | :--- | :--- | :--- |
> > > |Cora(Structure) | 90.83% |91.05% (+0.22%)| 88.71% (-2.12%)|
> > > |Texas(Label Leave-out) |85.22%|85.45% (+0.23%)| 82.13% (-3.09%)|
> > > |Ogbn-arxiv(Time Split) |75.49%|75.68% (+0.19%)| 71.35% (-4.14%)|
> > >
> > > The results from this comprehensive experiment are clear and consistent across all graph types:
> > > *   **Stability through Principled Objective:** When fine-tuning continues with our geometric objective, the ID-OOD separability is consistently maintained and slightly enhanced. The optimization process continues to enforce the beneficial angular structure, demonstrating that GeoEnergy provides a dynamically stable training methodology.
> > > *   **Degradation otherwise:** Conversely, reverting to a standard objective during updates leads to a degradation in performance. The carefully learned geometric structure is eroded. This is consistent with the well-known challenge of **catastrophic forgetting** in continual learning [1, 2] and is a general property of neural networks, not a specific flaw of our method.
> > >
> > > In essence, the key to maintaining ID-OOD separability in dynamic environments lies in the continued application of the geometric training principle.
> > >
> > > References:
> > > [1] McCloskey & Cohen, "Catastrophic Interference in Connectionist Networks," 1989.
> > >
> > > [2] Kirkpatrick et al., "Overcoming catastrophic forgetting in neural networks," PNAS 2017.
> > >
> > > ---
> > >
> > > Thanks for your invaluable feedback. We will incorporate the following major revisions into our final paper:
> > >
> > > 1.  **New experiments on three heterophilous datasets** to validate generalizability.
> > > 2.  **New comparisons with recent SOTA methods** (GRASP, GOLD, SPREAD) to better position our work.
> > > 3.  **A new Proposition 3 with a proof sketch** to provide deeper theoretical justification for improved OOD detection.
> > > 4.  **A new fine-tuning experiment** to demonstrate the dynamic stability of our method under model updates.
> > > 5.  **Clarifications** on the role of $s$ and our training methodology throughout the paper.
> > >
> > > We are confident these substantial revisions, motivated by your feedback, will significantly strengthen our paper.

---

> > > > ### Comment · Reviewer_D1EP · 2025-08-05
> > > >
> > > > Thank you for your effort in providing the additional explanations and experimental results. They have addressed my concerns and I will raise my score accordingly. I strongly encourage the authors to incorporate the clarifications and newly added comparative results into the revised version to improve the overall quality of the paper.

---

> > > > > ### Author Response · Authors · 2025-08-06
> > > > > **Thank you for the response!**
> > > > >
> > > > > We sincerely thank the reviewer for the positive feedback and for the increased score. We truly appreciate your recognition of our clarifications and additional experimental results. Following your suggestion, we will carefully incorporate the relevant explanations and comparative results into the revised version to enhance the clarity and overall quality of the paper. Your constructive comments have been very helpful in improving our work.

---

### Comment · Area_Chair_ptqt · 2025-08-04

Dear Reviewers,

As we near the end of the rebuttal period, this is a friendly reminder to submit your responses to the authors by **August 6**. Your engagement is crucial for our final decision.

**Action Required**

- Read the rebuttal carefully: Authors have invested significant effort in addressing your concerns.

- Reply directly to authors: Briefly acknowledge their points and indicate whether your assessment has changed (or why it remains unchanged).

- Update your review (if applicable): Adjust scores/comments in the review system to reflect your current stance.

Your AC

---

### Note · Authors · 2025-08-12

Dear Area Chair and Reviewers,

We thank all reviewers for the constructive review process. The discussion was highly valuable, and we are delighted by the positive outcomes. Specifically, we are grateful that Reviewer D1EP and 6itr will kindly raise their scores after we addressed their concerns. We also thank Reviewer FXpL for the insightful feedback that guided significant improvements to our paper. We are pleased that the reviewers consistently recognized the core strengths and novelty of our work, GeoEnergy:

*   **Novelty and Significance:** The work was praised for addressing a **relevant and under-explored issue** (R-D1EP) with a **geometrically principled approach** (R-6itr). Reviewer FXpL highlighted the novelty in **identifying logit coupling as a root cause** of OOD misclassification and noted that the paper's impact **extends beyond OOD detection** to improve calibration and self-training.

*   **Simplicity and Practicality:** All reviewers appreciated that the proposed solution is **simple, intuitive, and plug-and-play** (R-D1EP, R-6itr), making it easy to integrate into existing frameworks without costly retraining (R-FXpL).

Motivated by the excellent feedback, we have made substantial revisions to address all raised concerns. The final manuscript will be enriched with:

1.  Expanded Generalizability: New experiments on three heterophilous datasets confirming broad applicability.
2.  Stronger Empirical Validation: New comparisons with three recent SOTA methods (GRASP, GOLD, SPREAD), demonstrating competitive or superior performance. We also show GeoEnergy can further enhance SOTA methods like GRASP, highlighting its fundamental nature.
3.  Deeper Theoretical Grounding: A new Proposition 3 with a formal proof sketch, providing a direct theoretical link between our geometric decoupling and the improved ID-OOD energy separation.
4.  Comprehensive Ablations & Analyses: New ablation studies (justifying the preservation of the norm), an efficiency analysis (confirming negligible overhead), and a new fine-tuning experiment (demonstrating dynamic robustness).

We are confident that these extensive revisions, motivated by the reviewers' feedback, have significantly strengthened our paper. We believe it now presents a novel and effective solution to a key challenge in graph learning, supported by a more rigorous and comprehensive evaluation.

Thank you once again for your time and invaluable guidance.

Sincerely,

The Authors of Submission #14906

---

### Decision · Program_Chairs · 2025-09-17

**Decision:**

Accept (poster)

**Comment:**

This paper introduces GeoEnergy, a geometric logit decoupling framework for enhancing energy-based out-of-distribution (OOD) detection in graph neural networks (GNNs). The key idea is to **decouple the norm and direction of logits by normalizing class weights while preserving embedding norms as confidence signals**. This design leads to more structured energy distributions, sharper intra-class alignment, and improved calibration. By doing so, GeoEnergy is simple, plug-and-play, and compatible with existing GNNs, requiring no retraining or architectural changes.

The paper’s strengths lie in its clear geometric motivation, simplicity, and practicality. Reviewers agreed that it effectively addresses an underexplored challenge in graph OOD detection, offers a principled solution, and consistently improves detection performance across benchmarks. Furthermore, the proposed framework enhances confidence calibration and self-training, demonstrating broader applicability beyond OOD detection. The manuscript is also well written and easy to follow.

After the rebuttal, the authors successfully addressed most reviewer concerns. They added experiments on heterophilous and large-scale graphs, included direct comparisons with strong recent baselines (GRASP, GOLD, SPREAD), and strengthened the theoretical foundation with a new proposition. Efficiency analysis and ablation studies further justify the design choice of preserving embedding norms, addressing scalability, and design-related concerns.

In conclusion, this work makes a solid, well-validated, and practical contribution to graph OOD detection. I recommend acceptance, and encourage the authors to incorporate the reviewers’ suggestions into the camera-ready version to further strengthen the presentation.